# Targeting IL-17A enhances imatinib efficacy in Philadelphia chromosome-positive B-cell acute lymphoblastic leukemia

Feng Wang[1,2,8], Yunxuan Li[2,8], Zhaona Yang[1,3,4,8], Wenbin Cao[5,8], Ying Liu[2], Luyao Zhao[2], Tingting Zhang[2], Chenxi Zhao[2], Jinmei Yu[1,4], Jiaojiao Yu[1,4], Jichao Zhou[1,4], Xiaowei Zhang[1,4], Ping-Ping Li[1,4], Mingzhe Han[5], Sizhou Feng[5], Billy Wai-Lung Ng[6,7], Zhuo-Wei Hu[1,4], Erlie Jiang[5] ✉, Ke Li[2] ✉ & Bing Cui[1,4] ✉

Dysregulated hematopoietic niches remodeled by leukemia cells lead to imbalances in immunological mediators that support leukemogenesis and drug resistance. Targeting immune niches may ameliorate disease progression and tyrosine kinase inhibitor (TKI) resistance in Philadelphia chromosome-positive B-ALL (Ph[+] B-ALL). Here, we show that T helper type 17 (Th17) cells and IL-17A expression are distinctively elevated in Ph[+] B-ALL patients. IL-17A promotes the progression of Ph[+] B-ALL. Mechanistically, IL-17A activates BCR-ABL, IL6/JAK/STAT3, and NF-kB signalling pathways in Ph[+] B-ALL cells, resulting in robust cell proliferation and survival. In addition, IL-17A-activated Ph[+] B-ALL cells secrete the chemokine CXCL16, which in turn promotes Th17 differentiation, attracts Th17 cells and forms a positive feedback loop supporting leukemia progression. These data demonstrate an involvement of Th17 cells in Ph[+] B-ALL progression and suggest potential therapeutic options for Ph[+] B-ALL with Th17-enriched niches.

Philadelphia chromosome-positive B-cell acute lymphoblastic leukemia (Ph[+] B-ALL) represents a high-risk B-ALL subtype. Ph[+] B-ALL occurs in approximately 30% of young adult acute lymphoblastic leukemia (ALL) cases and accounts for more than 50% of ALL cases in patients older than 50 years of age[1]. With advances in allogeneic transplantation, chemotherapy, immunotherapy, and tyrosine kinase inhibitors (TKIs), including imatinib mesylate, dasatinib and ponatinib, the clinical outcomes of B-ALL patients have been significantly improved.

However, despite clinical improvements, a proportion of Ph[+] B-ALL patients still experience relapse, and managing relapse is still challenging[2]. Therefore, there is an urgent need to discover Ph[+] B-ALL drug targets or TKI combination strategies to combat Ph[+] B-ALL relapse. Recent evidence indicates that malignant cell-remodeled niche microenvironments play essential roles in the pathogenesis and chemoresistance of hematological malignancies[3]. The bone marrow (BM) niches remodeled by imbalances in immunological mediators

[1]State Key Laboratory of Bioactive Substance and Function of Natural Medicines, Institute of Materia Medica, Chinese Academy of Medical Sciences & Peking Union Medical College, 100050 Beijing, China. [2]Institute of Medicinal Biotechnology, Chinese Academy of Medical Sciences & Peking Union Medical College, 100050 Beijing, China. [3]Beijing Institute of Biological Products Company Limited, 100176 Beijing, China. [4]CAMS Key Laboratory of Molecular Mechanisms and Target Discovery of Metabolic Disorder and Tumorigenesis, Institute of Materia Medica, Chinese Academy of Medical Sciences & Peking Union Medical College, 100050 Beijing, China. [5]State Key Laboratory of Experimental Hematology, National Clinical Research Center for Blood Diseases, Haihe Laboratory of Cell Ecosystem, Institute of Hematology & Blood Diseases Hospital, Chinese Academy of Medical Sciences & Peking Union Medical College, 300020 Tianjin, China. [6]School of Pharmacy, Faculty of Medicine, The Chinese University of Hong Kong, Hong Kong, China. [7]Li Ka Shing Institute of Health Sciences, Faculty of Medicine, The Chinese University of Hong Kong, Hong Kong, China. [8]These authors contributed equally: Feng Wang, Yunxuan Li, Zhaona Yang, Wenbin Cao. ✉e-mail: jiangerlie@ihcams.ac.cn; like1986@163.com; cuibing@imm.ac.cn

activate survival pathways, protect against excessive ROS, induce metabolic reprogramming, and promote immunosuppression, consequently supporting leukemogenesis, cell proliferation and therapeutic resistance[4]. Thus, these leukemic niches provide an opportunity to identify therapeutic targets to improve the efficacy of TKI treatment in Ph[+] B-ALL[5,6].

T helper type 17 (Th17) lymphocytes secrete Th17-associated cytokines such as IL-17, IL-21, IL-22, and TNF and play crucial roles in autoimmune diseases, chronic inflammatory disorders, and cancer[7–11]. Experimental and clinical evidence suggests that IL-17A is a promising therapeutic target in many autoimmune and chronic inflammatory diseases[11,12]. Secukinumab and ixekizumab, two anti-IL-17A monoclonal antibodies, have been approved for the clinical treatment of psoriasis, psoriatic arthritis, and ankylosing spondylitis[13,14]. Recent research has indicated the roles of IL-17 in maintaining barrier integrity, establishing host defense under physiological conditions, and driving cancer progression under pathological conditions[15]. Previous studies have indicated that the increases in the Th17 cell population in B-cell ALL (B-ALL), acute myeloid leukemia (AML), and multiple myeloma (MM) are positively correlated with cancer progression and drug resistance[16–18]. Therefore, targeting Th17 cells may improve the outcomes of Ph[+] B-ALL treatment and reduce drug resistance. However, the exact roles of Th17 cells and Th17-associated cytokines in the Ph[+] B-ALL niche microenvironment and disease progression are still undefined.

The pathophysiological functions of Th17 cells and Th17-associated cytokines depend on the ability of IL-17 to induce proinflammatory mediators, the mitogenic effects in tissue progenitor cells and the ability to reprogram cellular metabolism[19]. In tumor microenvironments, the expression of immunological mediators is elevated by the inflammatory response in cancer cells, which plays vital roles in tumor metastasis, migration, proliferation and cancer stemness[5,20–25]. Chemokine (C-X-C motif) ligand 16 (CXCL16), a membrane-bound chemokine, acts as a ligand for C-X-C chemokine receptor type 6 (CXCR6) and contributes to the progression of many chronic inflammatory diseases, including fibrosis, nonalcoholic fatty liver disease, atherosclerosis, and cancer[26–31]. The expression of CXCR6 has been observed in Th17 cells, and the effector CD4+ cells with a CCR6[+]CXCR6[+] phenotype predominantly expressed classical Th17 markers such as IL-23R, IL-17A, and RORC. CCR6 and CXCR6 can be used to identify cytotoxic Th17 cells in experimental autoimmune encephalomyelitis[32]. CXCL16 can also act as a chemotactic agent for tumor-infiltrating T lymphocytes to create a microenvironment enhancing cancer progression[33,34]. However, the exact role of CXCL16 in Th17 cell differentiation and function, especially in the leukemia niche microenvironment, remains uncharacterized. Given that the Th17-associated cytokine IL-17A induces the production of various proinflammatory mediators, which can remodel the local microenvironment, we postulated that Th17 cells in the BM niche contribute to Ph[+] B-ALL development. We studied the functions and mechanisms of the niche-located inflammatory loop formed by Th17 cells, IL-17A, and CXCL16 in supporting Ph[+] ALL progression and illuminated potential therapeutic strategies.

Here, we show that Th17 cells and IL-17A are highly elevated in Ph[+] B-ALL patients, which in turn promote the progression of Ph[+] B-ALL by activating BCR-ABL, IL6/JAK/STAT3 and NF-kB signalling pathways and increasing the secretion of the chemokine CXCL16 by leukemia cells. CXCL16 further promotes Th17 cell differentiation, and recruitment and forms the positive loop in the niche microenvironment. As such, this study provides additional rationale for IL-17A- or CXCL16-directed therapy for patients with Ph[+] B-ALL disease.

## Results
### High IL-17A expression is associated with poor prognosis in patients with Ph[+] B-ALL
To investigate whether Th17 cells are enriched in Ph[+] B-ALL, we detected the frequency of Th17 cells in freshly isolated bone marrow mononuclear cells (BMMCs) from Ph[+] B-ALL patients and healthy donors (HDs). The frequency of Th17 cells was significantly higher in BMMCs from patients with Ph[+] B-ALL than in those from HDs (Fig. 1a and Supplementary Fig. 1a). However, the frequency of CD4[+] T cells among BMMCs from Ph[+] B-ALL patients was not different from that among BMMCs from HDs (Supplementary Fig. 1b). We then used a *p210 BCR/ABL*-inducible transgenic expression system[35]. Withdrawal of tetracycline administration in double transgenic mice (*BCR-ABL*[tTA]; C57BL/6 background) induced the expression of *BCR−ABL1* and resulted in the development of B-cell leukemia in 100% of the mice (Supplementary Fig. 1c). The survival time of *BCR-ABL*[tTA] mice after tetracycline withdrawal was as long as 15 weeks (Supplementary Fig. 1d). Necropsy demonstrated massive splenomegaly and enlargement of the majority of lymph nodes (LNs) in these mice (Supplementary Fig. 1e, f). Flow cytometric analysis demonstrated that the splenic lymphoblasts from the diseased mice were B220[dim] (Supplementary Fig. 1g, left) and expressed CD19, CD43, and BP-1 (Supplementary Fig. 1g, right). This pattern suggested that the transformed cells underwent arrest at the pre-B-cell stage of development and was similar to the cell surface marker expression pattern identified in Ph[+] B-ALL patient samples and *MMTV-BCR-ABL*[tTA] transgenic mice (FVB background)[36,37]. Additionally, the frequency of immature blasts was found to be increased in Wright-Giemsa-stained peripheral blood (PB) and HE-stained BM and spleen sections of *BCR-ABL*[tTA] mice compared to wild-type (*WT*) mice (Supplementary Fig. 1h). We then examined the frequency of Th17 cells in mice with BCR-ABL-driven B-ALL (*BCR-ABL*[tTA] mice). The results indicated that the frequency of Th17 cells among peripheral blood mononuclear cells (PBMCs) increased progressively during BCR-ABL-induced B-ALL development (Fig. 1b and Supplementary Fig. 1i), but the proportion of CD4[+] T cells in the PB of mice with BCR-ABL-driven B-ALL was similar to that in the PB of *WT* mice during B-ALL progression (Supplementary Fig. 1j). Moreover, the proportions of Th17 cells in the BM, spleen and LNs were elevated in *BCR-ABL*[tTA] mice compared to those in *WT* mice at the same age (Fig. 1c). Moreover, single-cell datasets from HDs and Ph[+] B-ALL patients (GSE134759) indicated that the proportion of Th17 cells among the BMMCs from patients with Ph[+] B-ALL was significantly higher than that among BMMCs from HDs (Fig. 1d, e). These data indicate that the frequency of Th17 cells is increased in Ph[+] B-ALL patients and the Ph[+] B-ALL-like mouse model.

Th17 cells can secrete various cytokines, including IL-17A, IL-17F, IL-21, and IL-22, which contribute to Th17-mediated diseases[9,10]. The mouse Th1/Th2/Th17 cytokine and chemokine assays showed increased concentrations of cytokines such as IL-17A, IL-5, IL-28, IL-17F in the plasma of *BCR-ABL*[tTA] mice, and IL-17A was among the top 5 of the 42 cytokines examined (Fig. 1f). The concentration of plasma IL-17A ranged from 100 pg/ml to 500 pg/ml in *BCR-ABL*[tTA] mice (Supplementary Fig. 1k). Th17 cells can also produce IL-21 and IL-22, and we then measured the concentrations of IL-21 and IL-22 in the plasma of *BCR-ABL*[tTA] mice and *WT* mice. Although a slight increasing trend in plasma IL-21 and IL-22 concentrations was observed in *BCR-ABL*[tTA] mice compared to *WT* mice, there was no statistically significant difference between these two groups (Supplementary Fig. 1l). It has been reported that IL-17A mediates signal transduction via the IL-17 receptor complex, comprising the IL-17RA and IL-17RC subunits[38]. Therefore, we queried *IL-17RA* and *IL-17RC* expression in Ph[+] B-ALL patients from the GEO database (GSE13204). Both *IL-17RA* and *IL-17 RC* were expressed in Ph[+] B-ALL patients. However, the mRNA expression of *IL-17RA* in Ph[+] B-ALL patients was lower than that in HDs (Fig. 1g). Consistently, protein expression results showed that IL-17A was significantly increased in BM from newly diagnosed Ph[+] B-ALL patients compared with that from HDs or Ph[-] B-ALL patients (Fig. 1h). Furthermore, we identified positive cell surface expression of IL-17RA and IL-17RC in primary Ph[+] B-ALL cells using flow cytometry analysis (Fig. 1i). Both IL-17RA and IL-17RC were detected in these cells by flow

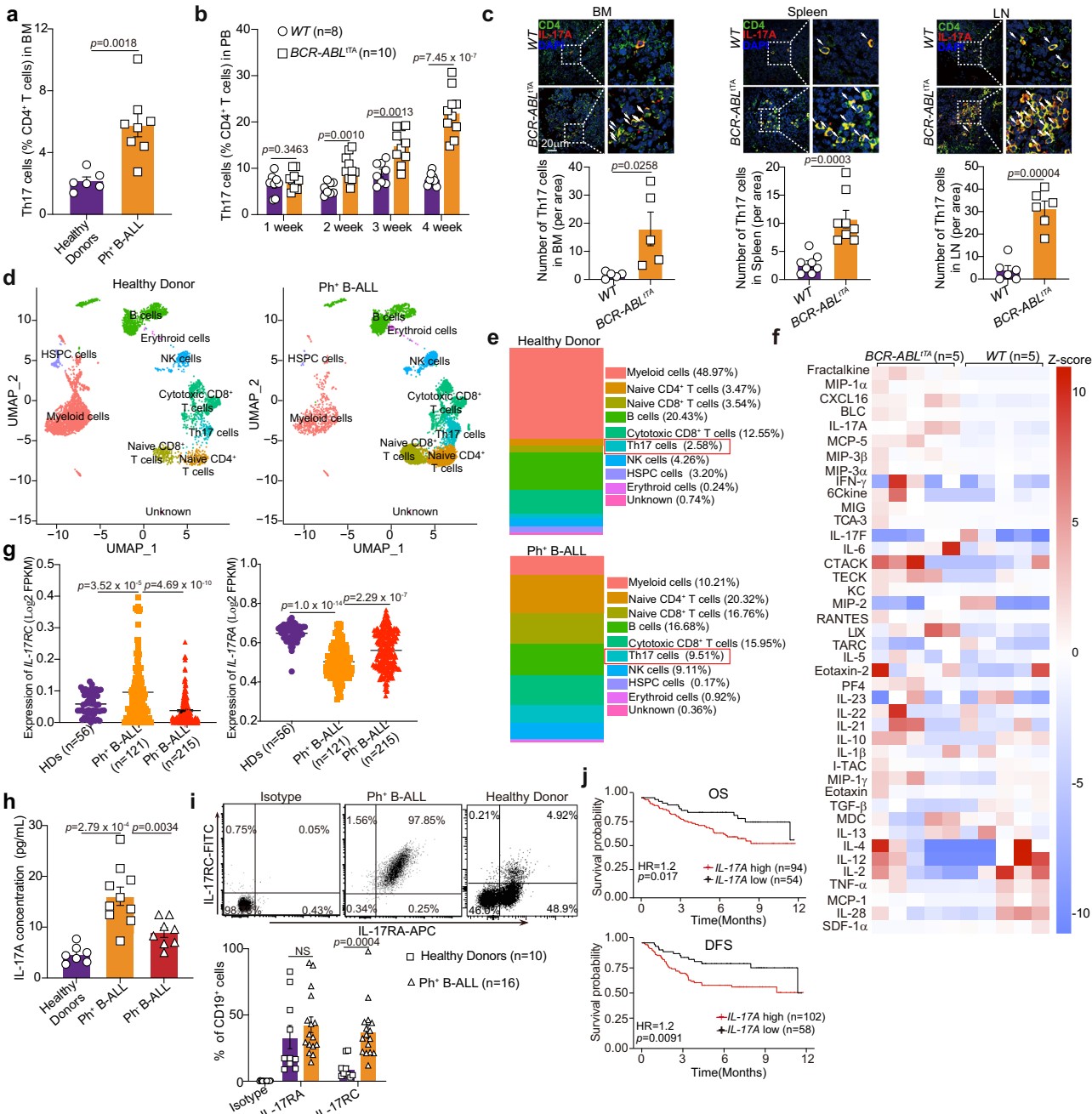

**Fig. 1 | IL-17A, the signature cytokine of Th17 cells, is positively associated with disease progression in Ph⁺ B-ALL. a** Flow cytometric analysis of the frequencies of Th17 cells within CD4⁺ T population in BMMCs from Ph⁺ B-ALL patients (*n* = 8 samples) or Healthy Donors (*n* = 6 samples). **b** Flow cytometric analysis of the frequencies of Th17 cells within the CD4⁺ T population in PBMCs of *WT* mice (*n* = 8 mice) or *BCR-ABL*ᵗᵀᴬ-transplanted mice (*n* = 10 mice) at the indicated times after transplantation. **c** Immunofluorescence analysis of the Th17 cells in the BM, spleen and LNs of *WT* mice and *BCR-ABL*ᵗᵀᴬ mice. *n* = 5 fields for BM, 8 fields for spleen, and 6 fields for LN from three different mice per group. **d, e** Uniform manifold approximation and projection (UMAP) visualization of different cell clusters (**d**) and stacked bar plot showing the percentages of indicated cells (**e**) in HD (*n* = 1 sample, GSM3732352) and Ph⁺ B-ALL patient (*n* = 1 sample, GSM3732354) from GSE134759 dataset. **f** Serum cytokine concentrations in *WT* mice and *BCR-ABL*ᵗᵀᴬ

mice were measured by Cytokines Array (*n* = 5 mice per group). Heatmaps with Z scores of serum cytokines are presented. **g** Log2 expression values for *IL-17RC* and *IL-17RA* in Ph⁺ B-ALL patients (*n* = 121 samples), Ph⁻ B-ALL patients (*n* = 215 samples), and HDs (*n* = 56 samples) were extracted from GEO database (GSE13204). **h** The IL-17A concentrations in the serum of Ph⁺ B-ALL patients (*n* = 10 samples), Ph⁻ B-ALL patients (*n* = 8 samples) and HDs (*n* = 7 samples) were measured by ELISA. **i** The expression of IL-17RA and IL-17RC on CD19⁺ cells from Ph⁺ B-ALL patients (*n* = 16 samples) and HDs (*n* = 10 samples) were detected by FSCs. **j** Kaplan–Meier survival curves of B-ALL patients stratified by *IL-17A* expression in B-ALL patients from the GSE11877 dataset. Statistical significance was calculated by (**a, b, c, i**) two-tailed Student's t-test; (**g, h**) one-way ANOVA with Tukey's multiple comparison tests; (**j**) two-sided log-rank test; Data are presented as means ± S.E.M. Source data are provided as a Source Data file.

cytometry. Interestingly, the expression of IL-17RC was significantly higher in Ph⁺ B-ALL cells than in normal B cells, whereas the expression of IL-17RA remained unchanged (Fig. 1i). In addition, patients with B-ALL expressing high *IL-17A* expression had significantly shorter

overall survival (OS) and disease-free survival (DFS) times than patients with B-ALL with low *IL-17A* expression (Fig. 1j). These data indicate that IL-17A secreted by Th17 cells positively correlates with Ph⁺ B-ALL progression.

## IL-17A promotes the proliferation, survival, and homing of Ph⁺ B-ALL cells

To determine whether IL-17A secreted by Th17 cells promotes leukemia development, we separately isolated Th17 cells and leukemia cells from Ph⁺ B-ALL patients and then cocultured the leukemia cells with the Th17 cells. The cocultured cells were treated with anti-human IL-17A neutralizing antibodies or human IgG1 (IgG1) for 24 h, and the proliferation and apoptosis activity of the leukemia cells were then evaluated (Fig. 2a). Coculture with Th17 cells increased the percentage of the Ki67⁺ leukemia subpopulation and the survival of Ph⁺ B-ALL cells, and these changes were abolished by treatment with the anti-IL-17A neutralizing antibody (anti-IL-17A) (Fig. 2b, c). Furthermore, we evaluated the direct effects of IL-17A on Ph⁺ B-ALL cell growth and survival in vitro. IL-17A alone significantly induced both dose- and time-dependent proliferation of SupB15 cells and BV173 cells (Fig. 2d, e). In addition, IL-17A maintained the survival (Fig. 2f) and proliferation activity (Supplementary Fig. 2a) of primary Ph⁺ B-ALL cells. Then, we performed homing assays (prior to 18 h) to investigate the effects of IL-17A treatment on the homing of transplanted leukemia cells (GFP-luc-tagged SupB15 cells). Flow cytometric analysis showed that IL-17A treatment enhanced the homing of leukemia cells to the BM and spleen in recipient mice (Supplementary Fig. 2b, c). To further explore the effect of rhIL-17A treatment on the engraftment of B-ALL cells, we performed rhIL-17A treatment 4 days after SupB15 cell transplantation (Fig. 2g). Biophotonic imaging on day 1 confirmed the successful transplantation of leukemia cells in all mice, with comparable leukemia burdens observed (Fig. 2h). By day 15, all mice transplanted with leukemia cells developed leukemia, indicating successful engraftment (Fig. 2h). The engraftment results showed that IL-17A treatment promoted Ph⁺ B-ALL progression (Fig. 2h and Supplementary Fig. 2d), as indicated by the increased infiltration of leukemia cells in the PB, BM, and spleen (Fig. 2i, j and Supplementary Fig. 2e), ultimately leading to a decreased overall survival rate of leukemia-engrafted mice (Fig. 2k). These data suggest that IL-17A plays a critical role in Ph⁺ B-ALL pathogenesis.

## IL-17A deficiency or neutralization attenuates the progression of Ph⁺ B-ALL

To further examine the in vivo role of IL-17A in leukemogenesis, we transplanted B-ALL cells isolated from *BCR-ABL*ᵗᵀᴬ mice into *WT* littermate mice and *IL-17A*-knockout (*IL-17A⁻/⁻*) mice. Over time, we monitored leukemia progression by detecting B220ᵈⁱᵐ CD19⁺ cells in PB from engrafted mice (Fig. 3a and Supplementary Fig. 3a). IL-17A deficiency significantly reduced the percentage of B220ᵈⁱᵐ CD19⁺ cells in the PB over time (Fig. 3b). Three weeks after transplantation, a substantial reduction in the immature blast cell population was observed in the PB and spleens of engrafted mice on the *IL-17A⁻/⁻* background (Fig. 3c). The spleen weights (Fig. 3d, e) and leukemia cell infiltration in the BM, spleen, and LNs (Fig. 3f) were decreased in *IL-17A⁻/⁻* recipients. In addition, IL-17A deficiency reduced the proportion of B220ᵈⁱᵐ CD19⁺ Ki-67⁺ cells in multiple organs and among PBMCs of engrafted mice (Fig. 3g and Supplementary Fig. 3b). Moreover, we performed homing assays in *IL-17A⁻/⁻* mice to detect the effect of *IL-17A* knockout on the homing of leukemia cells by using CFSE-labeled leukemia cells isolated from *BCR-ABL*ᵗᵀᴬ mice. We examined the distribution of leukemia cells in the BM and spleen of both *WT* and *IL-17A⁻/⁻* mice 16 h after transplantation. Flow cytometric analysis showed that *IL-17A* knockout reduced the homing of leukemia cells to the BM and spleen in recipient mice (Supplementary Fig. 3c).

Considering the blocking effects of anti-IL17A neutralizing antibodies, we first performed a homing assay to determine the impact of anti-IL-17A treatment on the homing of leukemia cells. We found that anti-IL-17A treatment reduced the homing of leukemia cells to the BM and spleen of recipient mice (Supplementary Fig. 3d). Then, we evaluated the therapeutic effect of anti-IL-17A or mouse IgG1 (IgG1) in B-ALL cell-engrafted mice (Fig. 3h). Twenty-one days after treatment, the mice

in the anti-IL-17A-treated group exhibited dramatic reductions in the spleen weight (Fig. 3i, j) and immature blast cell populations in the spleen and PBMCs (Fig. 3k). Similar to IL-17A deficiency, anti-IL-17A treatment reduced the number of B220ᵈⁱᵐ CD19⁺ cells (Fig. 3l) and Ki-67⁺ leukemia cells (Fig. 3m) in the BM, spleens, LNs, and PB of recipients, leading to a decrease in the death of B-ALL mice (Fig. 3n). More strikingly, the combination of anti-IL-17A antibodies and imatinib synergistically decreased leukemia cell infiltration and population in the spleens, PB, BM, and LNs of recipients and significantly enhanced the survival of B-ALL cell-engrafted mice (Fig. 3l-n), suggesting that combination treatment with anti-IL-17A and imatinib increased the therapeutic efficacy of imatinib. Overall, these data indicate that IL-17A depletion attenuates the progression of Ph⁺ B-ALL and exerts synergistic effects with imatinib to inhibit Ph⁺ B-ALL development.

## IL-17A activates the BCR-ABL signaling pathway to promote the proliferation of Ph⁺ B-ALL cells

IL-17A has pleiotropic effects on multiple target cells[39,40]. To understand the roles and potential molecular mechanisms of IL-17A in Ph⁺ B-ALL cells, we performed RNA sequencing (RNA-seq) and analyzed the gene expression profiles of primary mouse Ph⁺ B-ALL cells with or without recombinant mouse IL-17A (rmIL-17A) treatment (Fig. 4a). Gene set enrichment analysis (GSEA) showed that genes involved in the hallmark IL6/JAK/STAT3 signaling, hallmark TNF signaling via NF-kB, and hallmark inflammatory response pathways were highly enriched in primary mouse Ph⁺ B-ALL cells treated with IL-17A (Fig. 4b and Supplementary Fig. 4a, b). Moreover, we queried the PubMed GEO database from the MILE study (GSE13204) incorporating 575 B-ALL patients and 122 Ph⁺ B-ALL patients as previously described[41]. Patients with Ph⁺ B-ALL with *IL-17A* mRNA expression (208402_at) above the median level (*IL-17A high*) exhibited positive NES but no enrichment of the inflammatory response, IL6/JAK/STAT3 signaling and hallmark TNF signaling via NF-kB pathways compared with patients with leukemia with *IL-17A* mRNA expression below the median level (*IL-17A low*) (Supplementary Fig. 4c, d). Considering that leukemia cells acted mainly as responders but did not release IL-17A, IL-17RA, the receptor for IL-17A, should be a better marker of leukemia cells responding to IL-17A stimulation. We then conducted GSEA to compare patients with leukemia with *IL-17RA* mRNA expression (205707_at) above the median level (*IL-17RA high*) with patients with leukemia with *IL-17RA* mRNA expression below the median level (*IL-17RA low*). Similar to the findings in mouse primary B-ALL cells treated with IL-17A, the hallmark inflammatory response, hallmark IL6/JAK/STAT3 signaling and hallmark TNF signaling via NF-kB pathways were significantly enriched (FDR q < 0.25) in the *IL-17RA high* subgroup (Fig. 4c and Supplementary Fig. 4e). Additionally, we analyzed the published single-cell datasets from HDs and Ph⁺ B-ALL patients (GSE134759) and found that the hallmark TNF signaling via NF-kB pathway was enriched in B cells from Ph⁺ B-ALL patients compared to B cells from HDs (Fig. 4d). Furthermore, the BCR-ABL signaling pathway was indeed activated in the *IL-17A high* subgroup and *IL-17RA high* subgroups of Ph⁺ B-ALL patients (Supplementary Fig. 4f, g). We then conducted real-time PCR to monitor the major regulated genes involved in the above-mentioned signaling pathways. IL-17A treatment increased the transcription of *IL-6* and *Jak2* in primary mouse B-ALL cells (Fig. 4e). Moreover, IL-17A treatment increased IL-6 production in SupB15 B-ALL cells (Fig. 4f) and primary Ph⁺ B-ALL cells (Fig. 4g). In fact, the concentration of plasma IL-17A ranged from 100 pg/ml to 500 pg/ml in *BCR-ABL*ᵗᵀᴬ mice and from 10 pg/ml to 50 pg/ml in patients with B-ALL (Supplementary Fig. 1j, g). We then investigated the effect of pathological concentrations of IL-17A on the transcription of *IL-6* and *JAK2* in B-ALL cells. Treatment with 200 pg/ml IL-17A resulted in increases in the mRNA levels of *Il-6* and *Jak2*, whereas treatment with 20-100 pg/ml IL-17A did not result in this effect (Fig. 4h). Additionally, we observed that treatment with pathological concentrations of IL-17A (200-500 pg/ml)

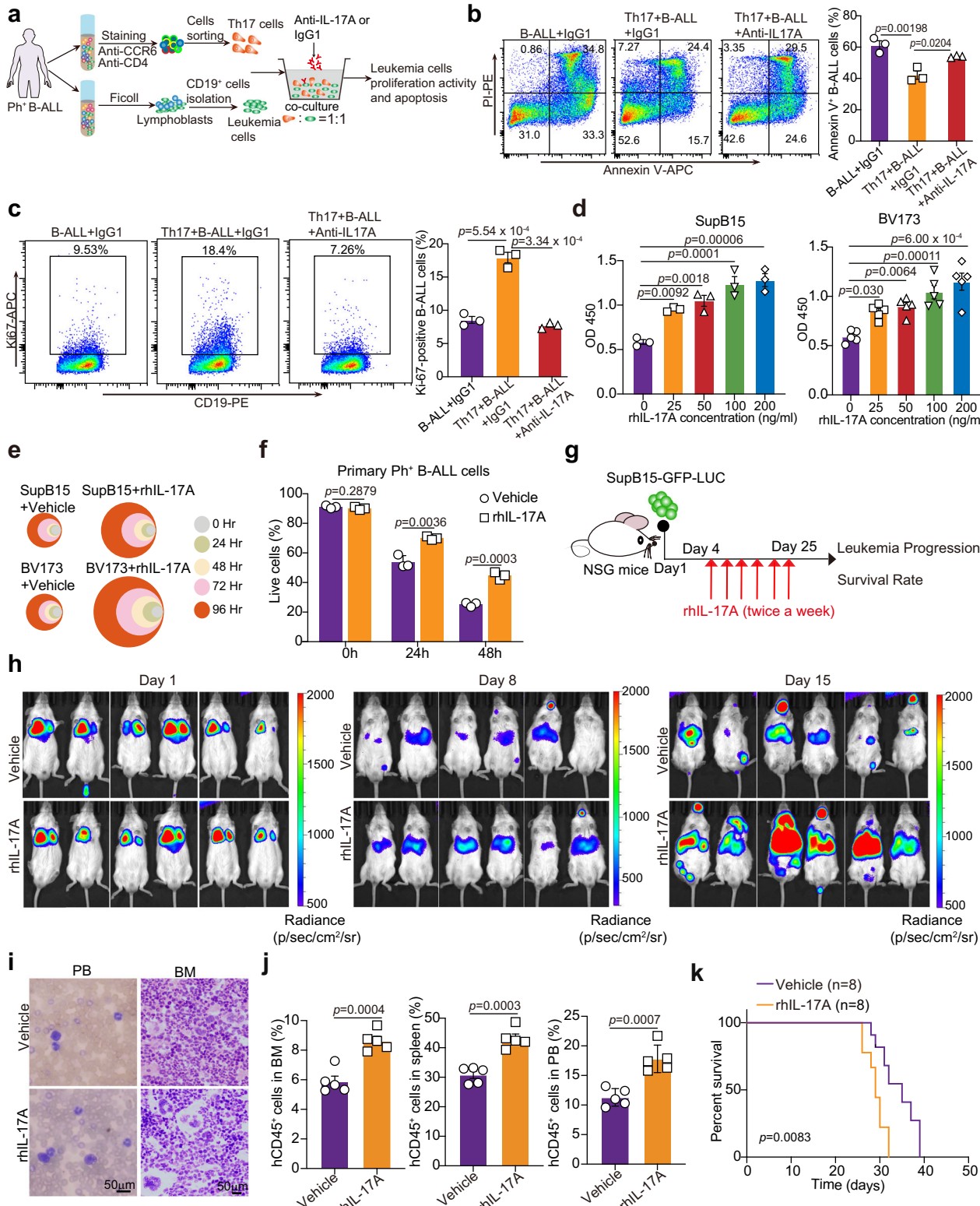

significantly increased the mRNA levels of *Il-6* and *Jak2* (Fig. 4h). Consistently, treatment with pathological concentrations of IL-17A (200–500 pg/ml) increased the phosphorylation of BCR-ABL, STAT5, AKT, STAT3, p65 and MEK1/2 (Fig. 4i). These results suggest that IL-17A can activate the BCR-ABL, IL6/JAK/STAT3, and NF-kB signaling pathways in B-ALL cells.

### CXCL16 secreted from leukemia cells stimulated with IL-17A promotes the differentiation and migration of Th17 cells

The crosstalk between B-ALL cells and the surrounding microenvironment in BM creates a BM malignant niche and accelerates disease progression[42,43]. Indeed, we found that Th17 cells accumulated within the B-ALL BM microenvironment niche. Moreover, these Th17 cells are

**Fig. 2 | IL-17A promotes the proliferation and survival of Ph⁺ B-ALL cells.**
**a** Schematic experimental design showing Th17 cells cocultured with leukemia cells with or without anti-IL-17A mAb treatment. **b, c** Representative FACS plots and flow cytometric analysis of leukemia cell apoptosis as determined by Annexin-V and PI staining (**b**) and Ki-67⁺ leukemia cells (**c**) in the coculture system with or without anti-IL-17A mAb treatment. **b, c** $n = 3$ independent experiments. The strategy for isolating Th17 cells is shown in Supplementary Fig. 6a. **d** SupB15 ($n = 3$ independent experiments) and BV173 cells ($n = 5$ independent experiments) incubated with different concentrations of rhIL-17A were detected by CCK-8 assay. **e** Relative cell proliferation viabilities of SupB15 cells and BV173 cells incubated with rhIL-17A (100 ng/ml) or vehicle were detected by CCK-8 assay at the indicated times. The colors represent different time points, and the diameter indicates the relative cell proliferation (normalized to the baseline on day 1). ($n = 4$ independent experiments) **f** The viability of primary Ph⁺ B-ALL cells incubated with or without rhIL-17A (100 ng/ml) was determined by a cell counter ($n = 3$ independent experiments).

**g** Schematic representation of the SupB15-GFP-Luc cell xenograft experiment. **h** The infiltration of B-ALL cells in vivo was quantitatively monitored by a biophotonic imaging system in mice treated with rhIL-17A or vehicle. The infiltration of B-ALL cells was monitored on days 1, 8, and 15 after transplantation and representative images of indicated groups were shown ($n = 6$ mice per group). **i** Representative images of Wright-Giemsa-stained PB smears and H&E staining of spleens from the indicated groups ($n = 3$ mice per group). **j** Flow cytometric analysis of the percentage of hCD45⁺ cells in the PB, spleen and BM of indicated groups ($n = 5$ mice per group). The gating strategy for hCD45⁺ cells is shown in Supplementary Fig. 2e. **k** Kaplan–Meier survival curves of SupB15-GFP-LUC cell-engrafted NSG mice treated with rhIL-17 or vehicle ($n = 8$ mice per group). Statistical significance was calculated by (**b, c, d**) one-way ANOVA with Tukey's multiple comparison tests; (**f, j**) two-tailed Student's t-test; (**k**) two-sided log-rank test. Data are presented as means ± S.E.M. Source data are provided as a Source Data file.

located in the surroundings of Ph⁺ B-ALL cells (Fig. 5a). Therefore, we then investigated whether Th17 cells can be recruited by Ph⁺ B-ALL cells by detecting the recruitment of Th17 cells in a noncontact coculture system with Ph⁺ B-ALL cells. Leukemia cells triggered the migration of Th17 cells, which was further promoted by rhIL-17A stimulation (Fig. 5b). Chemokines are essential factors in attracting specific cell types to the tumor microenvironment[44]. To investigate which chemokine participates in the migration of Th17 cells to leukemia cells, we detected the effect of IL-17A on the chemokine secretion of leukemia cells. The expression of *CXCL5*, *IL-5*, *CXCL16*, and *CCL5* was significantly increased in both SupB15 and BV173 cells after rhIL-17A treatment (Fig. 5c). We also detected the effect of IL-17A on chemokine expression in a Ph⁻ leukemia cell line (NALM-6). The expression levels of *CXCL16*, *CCL5*, and IL-5 increased after rhIL-17A treatment (Supplementary Fig. 5a). These results align with our observations in the Ph⁺ leukemia cell lines SupB15 and BV173 (Fig. 5c). However, the fold change in *CXCL16* expression in NALM-6 cells treated with IL-17A (1-3-fold) was comparatively lower than those observed in SupB15 (9-14-fold) and BV173 (4-5-fold) cells.

We then assessed the impact of IL-17A on the secretion of CXCL5, IL-5, CXCL16, and CCL5. Our findings revealed significant increases in the secretion of IL-5, CXCL5, CCL5, and CXCL16 in SupB15 cells following rhIL-17A treatment, and among these four cytokines, CXCL16 exhibited the most robust secretion from Ph⁺ B-ALL cells (Fig. 5d). In addition, the plasma concentration of CXCL16 in *BCR-ABL*^tTA mice (100–300 pg/ml) was increased at least 8-fold compared to that in *WT* mice (10–50 pg/ml) (Fig. 5e). Moreover, the serum CXCL16 concentration was significantly increased in newly diagnosed Ph⁺ B-ALL patients compared with HDs (Fig. 5f). Furthermore, we quantified the systemic CXCL16 level in *BCR-ABL*^tTA mice treated with IgG or anti-IL-17A. Compared to IgG treatment, anti-IL-17A treatment significantly reduced the serum CXCL16 concentration (Supplementary Fig. 5b). Regarding Th17 cell functions, we then investigated the effects of CXCL16 secreted from Ph⁺ B-ALL cells on Th17 cells in vitro. Naïve CD4⁺ T cells were isolated and induced to differentiate into Th17 cells in the presence or absence of CXCL16 (Fig. 5g). CXCL16 significantly triggered the differentiation of Th17 cells in a dose-dependent manner (Fig. 5h) but had no effect on the proliferation activity of Th17 cells (Supplementary Fig. 5c). Because the level of CXCL16 in the plasma of *BCR-ABL*^tTA mice was 100–300 pg/ml, we then performed a Th17 cell in vitro differentiation assay to investigate whether treatment with 200 pg/ml CXCL16 could affect the differentiation of naïve CD4⁺ T cells into Th17 cells. We found that treatment with 200 pg/ml CXCL16 induced the differentiation of naïve CD4⁺ T cells into Th17 cells (Supplementary Fig. 5d). To determine whether CXCL16 secreted by leukemia cells promotes the migration of Th17 cells, which in turn maintains the survival of leukemia cells, we separately isolated Th17 cells and leukemia cells from Ph⁺ B-ALL patients and then cocultured the leukemia cells with the Th17 cells in a noncontact coculture system (Fig. 5i). Treatment with an anti-CXCL16 neutralizing antibody (anti-CXCL16) significantly inhibited the migration of Th17

cells (Fig. 5j) and the proliferation activity of leukemia cells (Fig. 5k). These data indicate that CXCL16 secreted from leukemia cells promotes the differentiation and migration of Th17 cells, which support leukemia cell proliferation.

We then investigated how IL-17A stimulated the secretion of CXCL16 from leukemia cells. IL-17A increased the expression levels of both *CXCL16* mRNA and protein in Ph⁺ B-ALL cells (Fig. 6a, b). In the primary B-ALL mouse model, we also found that CXCL16 accumulated more in the spleen niche in *BCR-ABL*^tTA mice than in WT mice (Fig. 6c). Moreover, rmIL-17A treatment further increased CXCL16⁺ cell infiltration in the spleens of *BCR-ABL*^tTA mice compared to WT mice (Fig. 6c). To exclude the possibility that the increase in CXCL16 accumulation in the spleens of B-ALL mice was due to the high leukemia burden, we measured CXCL16 expression on a per-cell basis in primary B-ALL cells after rmIL-17A treatment by FACS. Flow cytometric analysis revealed an increase in the mean fluorescence intensity of CXCL16 in Ph⁺ B-ALL cells after IL-17A treatment, suggesting that IL-17A increases CXCL16 expression in Ph⁺ B-ALL cells on a per-cell basis (Fig. 6d). As indicated in primary mouse and human B-ALL cells (Fig. 4b and Supplementary Fig. 4d), IL-17A activated the NF-κB signaling pathway, which plays a critical role in inducing the expression and secretion of chemokines and cytokines[45]. Our results indicated that rmIL-17A significantly increased p65 phosphorylation in the nucleus in a dose-dependent manner in primary mouse B-ALL cells (Fig. 6e). Immunofluorescence staining also indicated elevated phosphorylated activity of p65 upon rhIL-17A treatment (Fig. 6f). Furthermore, NF-κB reporter luciferase activity was increased in a concentration-dependent manner at 24 h following rhIL-17A treatment (Fig. 6g). These results indicate that IL-17A increases NF-κB transcriptional activity in primary B-ALL cells.

To further investigate whether CXCL16 is the target gene of NF-κB, we performed chromatin immunoprecipitation (ChIP)-qPCR analyses with the truncated promoter region of *CXCL16* (Fig. 6h, top). We found that NF-κB could bind to sequences from -2000 bp to -1314 bp and from −424 bp to −178 bp within the CXCL16 putative promoter region and that rmIL-17A increased the binding activity of NF-κB to the CXCL16 promoter (Fig. 6h, bottom). To further confirm whether IL-17A stimulates the secretion of CXCL16 through NF-κB activation, we applied the IKK inhibitor BAY-117082 to inhibit NF-κB activation (Fig. 6i) and measured CXCL16 expression after treatment. Notably, BAY-117082 inhibited *CXCL16* mRNA and CXCL16 protein expression, but rmIL-17A treatment only weakly restored *CXCL16* mRNA and CXCL16 protein expression (Fig. 6j, k), suggesting that NF-κB activation is required for IL-17A-induced CXCL16 expression. These results indicate that rmIL-17A increases CXCL16 secretion in a manner dependent on NF-κB activation in primary B-ALL cells.

### CXCL16 depletion attenuates the progression of Ph⁺ B-ALL

We then investigated the leukemogenesis role of CXCL16 in vivo. After secondary transplantation of *BCR-ABL*^tTA B-ALL cells, the transplanted

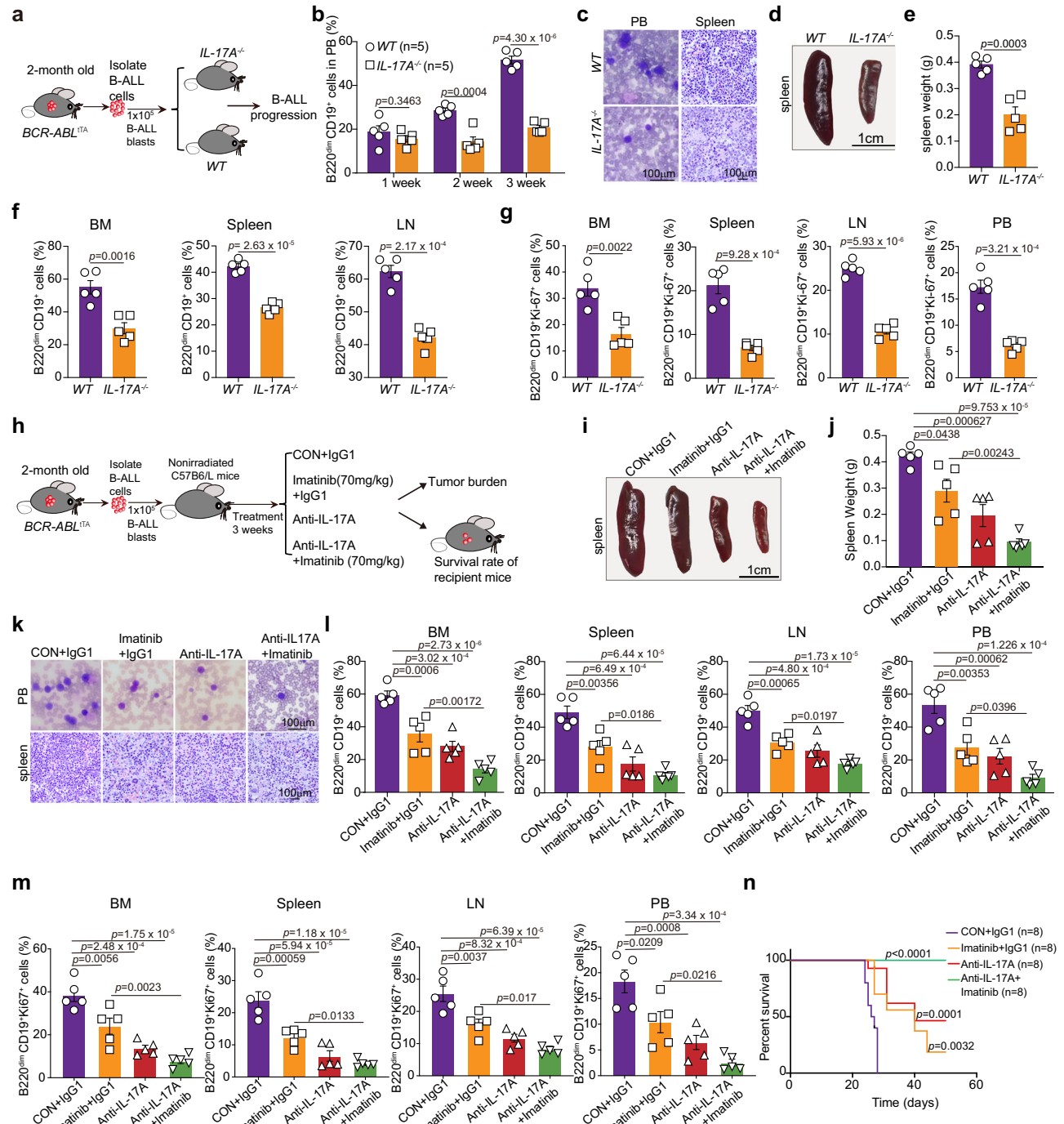

**Fig. 3 | IL-17A deficiency or neutralization attenuates the progression of Ph⁺ B-ALL. a** Schematic strategy for investigating the role of IL-17A knockout in leukemogenesis. **b** Flow cytometric analysis of the percentage of B220$^{dim}$ CD19$^+$ cells in the PB of *WT* mice and *IL-17A$^{-/-}$* mice with secondary transplantation at the indicated times (*n* = 5 mice per group). **c–e** Representative images of Wright-Giemsa-stained PB smears (**c, left**), H&E staining of the spleen (**c, right**), spleen (**d**) and spleen weight (**e**) in *WT* or *IL-17A$^{-/-}$* mice with secondary transplantation (*n* = 5 mice per group). **f, g** Flow cytometric analysis of the percentage of B220$^{dim}$ CD19$^+$ cells (**f**) and B220$^{dim}$ CD19$^+$ Ki-67$^+$ cells (**g**) in the BM, spleen, LNs and PBMCs from *WT* mice and *IL-17A$^{-/-}$* mice with secondary transplantation (*n* = 5 mice per group). **h** Strategy for investigating the effects of anti-IL-17A mAb alone or combined with imatinib on Ph⁺ B-ALL progression in vivo. **i–k** Representative images of spleens (**i**), statistical analysis of the spleen weight (**j**), representative images of Wright-Giemsa-stained

PB smears (**k, top**) and H&E staining of spleens (**k, bottom**) from mice with secondary transplantation treated with the indicated agents (*n* = 5 mice per group). **l–m** Flow cytometric analysis of the percentages of B220$^{dim}$ CD19$^+$ cells (**l**) and B220$^{dim}$ CD19$^+$ Ki-67$^+$ cells (**m**) in the PB, BM, LNs and spleens of mice with secondary transplantation treated with the indicated agents (*n* = 5 mice per group). **n** Kaplan–Meier survival curves of mice with secondary transplantation treated with the indicated agents (*n* = 8 mice per group). **b, f, l** The gating strategy for B220$^{dim}$ CD19$^+$ cells was shown in Supplementary Fig. 3a. **g, m** The gating strategy for B220$^{dim}$ CD19$^+$ Ki67$^+$ cells was shown in Supplementary Fig. 3b. Statistical significance was calculated by (**b, e, f, g**) two-tailed Student's t-test; (**j, l, m**) one-way ANOVA with Tukey's multiple comparison tests. (**n**) two-sided log-rank test; Data are presented as means ± S.E.M. Source data are provided as a Source Data file.

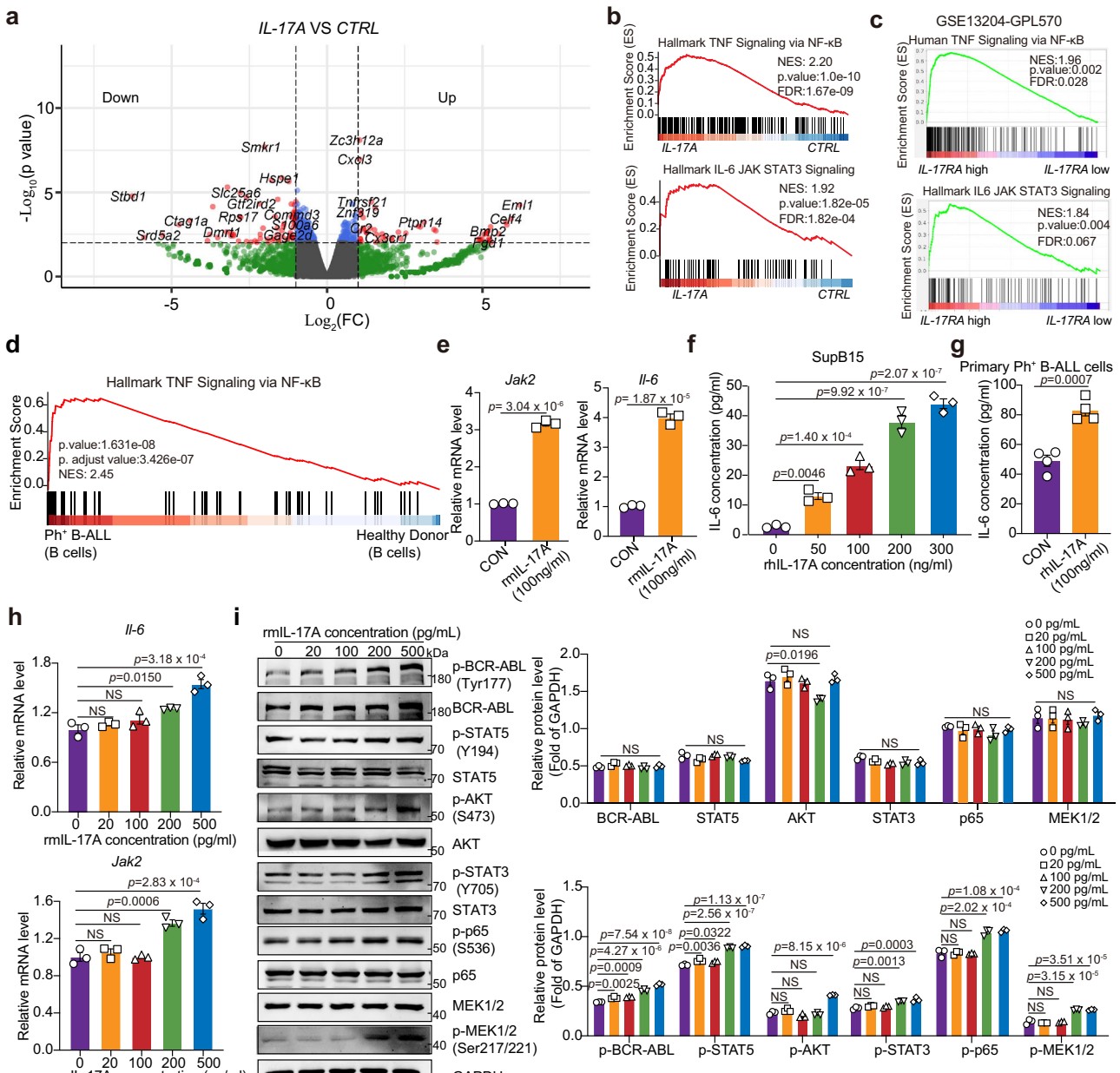

**Fig. 4 | IL-17A activates the BCR-ABL and JAK/STAT3 signaling pathways to promote the proliferation of Ph⁺ B-ALL cells. a** Volcano plot of differentially expressed genes in primary mouse B-ALL cells with or without rmIL-17A treatment. According to the criteria of Log2(fold-change) >1.5 or <-1.5 and *P* value < 0.05, selected upregulated and downregulated genes are highlighted in the volcano plot. **b** Molecular signature classification screening by GSEA with the top enriched signaling pathways in primary mouse leukemia cells treated with or without rmIL-17A. **a**, **b** *n* = 3 independent experiments. **c** GSEA demonstrating the enrichment of gene sets in Ph⁺ B-ALL patients with *IL-17RA* mRNA ('205707_at) expression above the median level (*IL-17RA* high) (*n* = 61) and below the median level (*IL-17RA* low) (*n* = 61). **d** GSEA demonstrating the enrichment of indicated gene set in B cells from Ph⁺ B-ALL patients and HDs in GSE134759 dataset. **e** Real-time PCR analysis of the relative mRNA levels of *Jak2* and *Il-6* in primary mouse leukemia cells treated with or without rmIL-17A (*n* = 3 independent experiments). **f**, **g** The human IL-6

concentrations in the culture supernatant of SupB15 cells (*n* = 3 independent experiments) (**f**) and primary Ph⁺ B-ALL cells (**g**) treated with different concentrations of rhIL-17A were measured by ELISA (*n* = 4 independent experiments). **h** Real-time PCR analysis of the relative mRNA levels of *Jak2* and *Il-6* in primary mouse leukemia cells treated with pathological concentrations of rmIL-17A (*n* = 3 independent experiments). **i** The levels of proteins in primary mouse leukemia cells treated with or without rmIL-17A were measured by Western blotting. *n* = 3 independent experiments. The samples were derived from the same experiment, and the gels/blots were processed in parallel. **a**–**d** Statistical significance was determined by a one-sided permutation test, and statistical adjustments were made for multiple comparisons. Statistical significance was calculated by (**e**, **g**) two-tailed Student's t-test; (**f**, **h**, **i**) one-way ANOVA with Tukey's multiple comparison tests; Data are presented as means ± S.E.M. Source data are provided as a Source Data file.

mice were treated with rmCXCL16 for 2 weeks. The percentages of B220^dim CD19⁺ cells in the BM, spleens, LNs and PB were significantly higher in rmCXCL16-treated Ph⁺ B-ALL syngeneic transplant mice than in PBS-treated mice (Fig. 7a). Indeed, the percentage of immature blast cells in the PB (Fig. 7b), leukemia cell infiltration in the spleen (Fig. 7c),

the spleen weight (Fig. 7d) and the percentage of Ki-67⁺ cells in the spleen (Fig. 7e) were substantially increased after rmCXCL16 treatment in mice with secondary *BCR-ABL^tTA* B-ALL cell transplantation. Furthermore, we determined the percentage of Th17 cells in the leukemia niche in mice with secondary *BCR-ABL^tTA* B-ALL cell transplantation

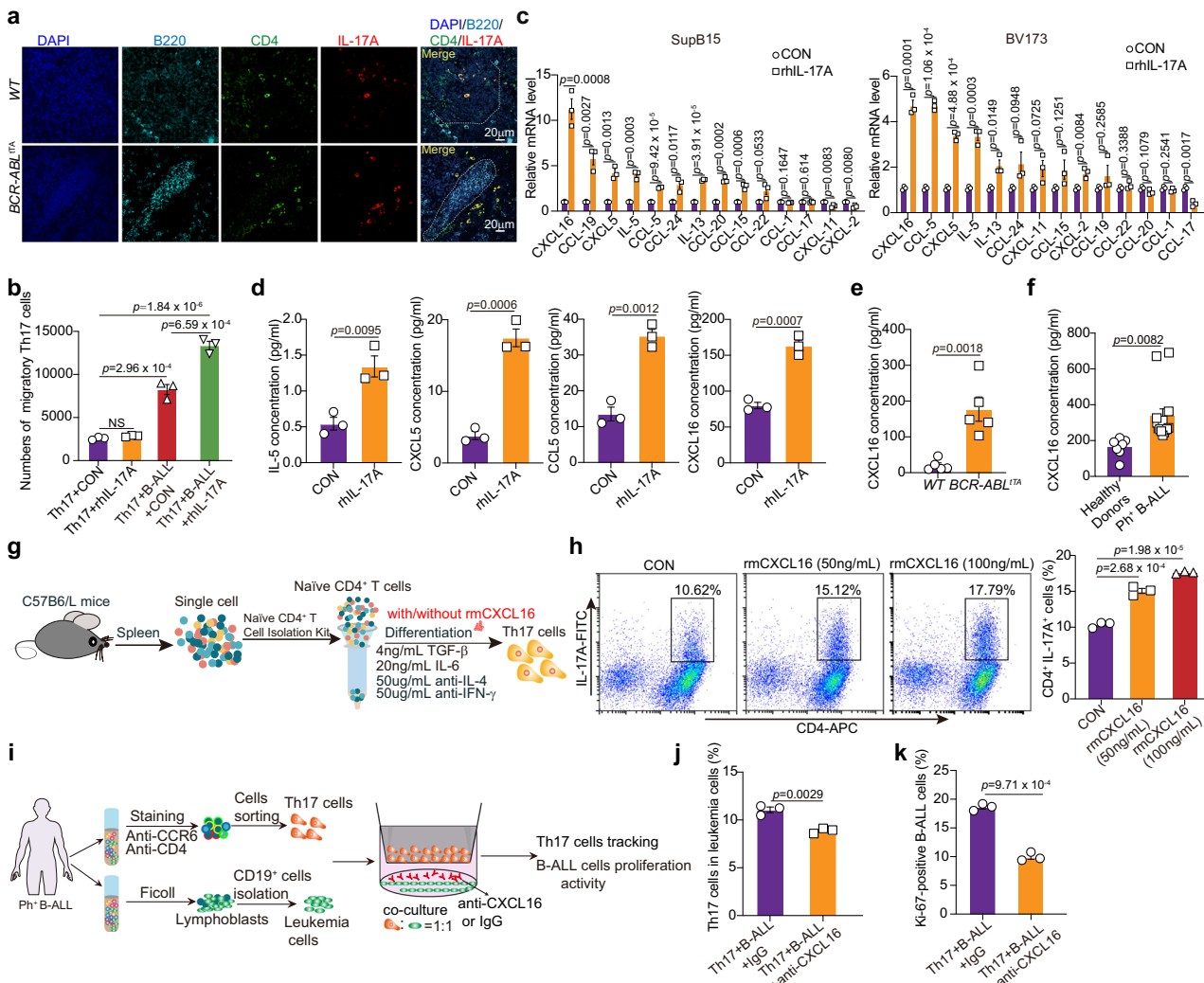

**Fig. 5 | IL-17A stimulates the secretion of CXCL16 from leukemia cells, promoting the differentiation and migration of Th17 cells. a** B220+ cells and Th17 cells in the BM niche in *WT* mice and *BCR-ABL^tTA* mice (*n* = 3 mice per group) were subjected to immunofluorescence staining. Representative images were shown. **b** The numbers of Th17 cells migrating to culture medium or to leukemia cells treated with or without 50 ng/ml rhIL-17A were determined by cell counting and FCS analysis. **c** Real-time PCR analysis of the relative mRNA levels of chemokine genes in SupB15 (left) and BV173 cells (right) treated with rhIL-17A (100 ng/ml) or PBS control. **d** The human IL-5, CXCL5, CCL5 and CXCL16 levels in the culture supernatant of SupB15 cells treated with or without rhIL-17A (100 ng/ml) were measured by ELISA. **e** The CXCL16 concentrations in the serum of *BCR-ABL^tTA* mice and *WT* mice (*n* = 5 mice per group) were quantified with a Mouse Chemokine Assay Q1 (Raybiotech). **f** The CXCL16 concentrations in the serum of HDs (*n* = 7 samples)

and primary Ph+ B-ALL patients (*n* = 14 samples) were measured by ELISA. **g** Schematic strategy for studying the differentiation of Th17 cells with or without rmCXCL16 stimulation in vitro. **h** Representative FACS plots and quantification of the percentage of Th17 cells in naïve CD4+ T cells with or without rmCXCL16 treatment. **i** Schematic strategy for studying the migration of Th17 cells cocultured with leukemia cells with or without anti-CXCL16 mAb treatment in vitro. **j, k** Flow cytometric analysis of the percentage of Th17 cells in the lower chambers of the inserts (**j**) and proliferation activity of leukemia cells (**k**) in a coculture system with or without anti-CXCL16 treatment. The gating strategy for isolating Th17 cells is shown in Supplementary Fig. 6a. (**b–d, h, j, k**) *n* = 3 independent experiments. Statistical significance was calculated by (**c–f, j, k**) two-tailed Student's t test; (**b, h**) one-way ANOVA with Tukey's multiple comparison tests; Data are presented as means ± S.E.M. Source data are provided as a Source Data file.

treated with or without CXCL16. CXCL16 dramatically increased the percentage of Th17 cells in the BM, spleen, LNs and PB (Fig. 7f, g and Supplementary Fig. 6b), indicating that CXCL16 promotes the migration of Th17 cells to the leukemia niche in vivo.

CXCL16 treatment accelerated the death of mice with secondary transplantation of *BCR-ABL^tTA* cells, indicating that CXCL16 promotes the progression of Ph+ B-ALL (Fig. 7h). We then examined the therapeutic effect of anti-CXCL16 or goat IgG isotype control (IgG) in the BCR-ABL-induced B-ALL secondary transplantation mouse model (Fig. 7i). On day 21 after treatment, anti-CXCL16 treatment reduced leukemia cell infiltration in the spleen (Fig. 7j); the spleen weight (Fig. 7k); the populations of immature blast cells in the PB and spleen (Fig. 7l); the percentages of B220^dim CD19+ cells in the PB, BM, spleen

and LNs (Fig. 7m); and the percentage of Ki-67+ cells in the spleen (Fig. 7n). Moreover, anti-CXCL16 treatment significantly reduced the percentages of Th17 cells in the BM, spleen, LNs and PB of B-ALL cell-transplanted mice, indicating that anti-CXCL16 treatment impeded the migration of Th17 cells to the leukemia niche in vivo (Fig. 7o). Additionally, combination treatment with anti-CXCL16 and imatinib further reduced leukemia progression and the percentage of Th17 cells in the leukemia niche in mice with secondary transplantation (Fig. 7j–o), suggesting that combination treatment with anti-CXCL16 and imatinib increased the therapeutic efficacy of imatinib. These data indicate that anti-CXCL16 treatment, by blocking Th17 cell activities in the BM niche, attenuates the progression of Ph+ B-ALL (Fig. 8).

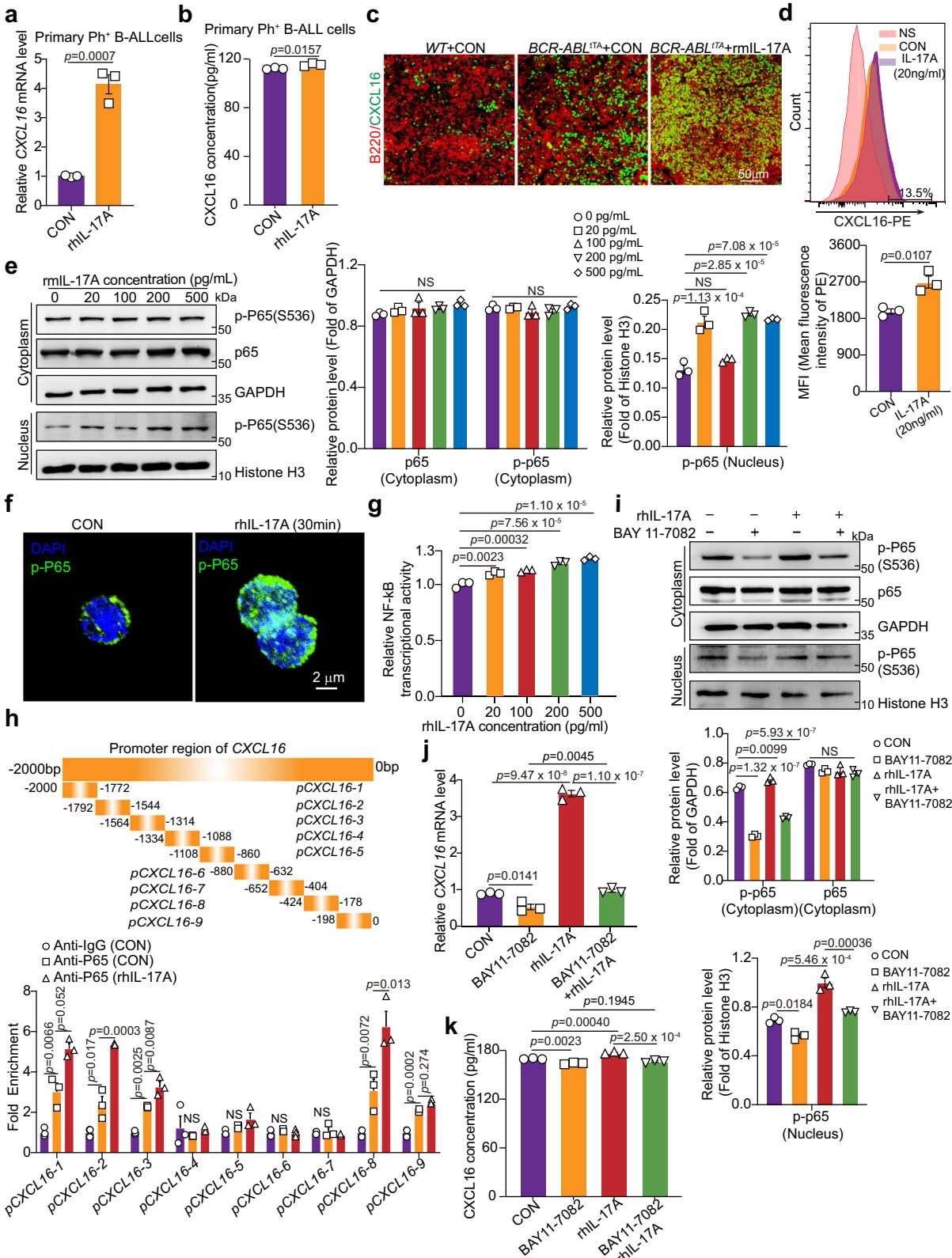

## Discussion

Th17 cells accumulate specifically in various tumors, indicating the occurrence of targeted recruitment of these cells by the tumor microenvironment. Th17 cells play a crucial role in inflammation and tumor immunity through the secretion of Th17-associated cytokines such as IL-17A, IL-21, IL-23, and TNF[46]. Studies have demonstrated that

hematological malignancies, including multiple myeloma, B-cell lymphoma, AML, and ALL, exhibit elevated frequencies of Th17 cells[16,17,47–49]. These increased numbers of Th17 cells, along with increased levels of IL-17 and other proinflammatory cytokines, contribute to growth, drug resistance, and apoptosis inhibition in B-cell lymphoma cells, as well as suppression of immune responses. These

**Fig. 6 | IL-17A activates NF-кB** to induce *CXCL16* transcription and **CXCL16 secretion in leukemia cells.** Primary Ph⁺ B-ALL cells were treated with or without rhIL-17A for 24 h, and relative *CXCL16* mRNA level (**a**) and CXCL16 secretion (**b**) were evaluated by real-time PCR and ELISA, respectively. **c** Representative immunofluorescence images of B220 (red) and CXCL16 (green) staining in spleen tissue sections of *WT* mice and *BCR-ABL*^tTA mice treated with or without 50 µg/kg rmIL-17A for a duration of 3 weeks (twice a week) (*n* = 3 mice per group). **d** Flow cytometric analysis and quantification of CXCL16 expression in primary mouse B-ALL cells treated with or without rmIL-17A. **e** The protein levels of p65 and p-p65 in the cytoplasm and p-p65 in the nucleus of primary mouse leukemia cells treated with or without rmIL-17A were measured by Western blotting.
**f** Immunofluorescence of p-P65 in primary B-ALL cells treated with or without rhIL-17A. **g** The effect of rhIL-17A treatment on NF-kB transcriptional activity. HEK 293T cells were transfected with a synthetic NF-kB luciferase reporter construct

(pNF-kB-Luc) for 12 h and then treated with different concentrations of rhIL-17A for 24 h. NF-kB transcriptional activity was detected by a luciferase assay. **h** ChIP–qPCR analyses of the binding of NF-kB to the CXCL16 promoter region in SupB15 cells treated with or without rhIL-17A. **i**–**k** The effect of BAY11-7082, rIL-17A or BAY11-7082 in combination with rIL-17A on the cytoplasmic and nuclear protein levels of p65, p-p65, and CXCL16 in primary Ph⁺ B-ALL cells. **i** The indicated protein band intensities were quantified using ImageJ software. The relative *CXCL16* mRNA level (**j**) in primary Ph⁺ B-ALL cells and CXCL16 protein level (**k**) in the supernatant of primary Ph⁺ B-ALL cells were measured by RT–PCR and ELISA, respectively. (**a**, **b**, **d**–**k**) *n* = 3 independent experiments. Statistical significance was calculated by (**a**, **b**, **d**) two-tailed Student's t-test; (**e**, **g**–**k**) one-way ANOVA with Tukey's multiple comparison tests; Data are presented as means ± S.E.M. Source data are provided as a Source Data file.

studies highlight the important roles of Th17 cells and IL-17A in leukemia. However, determining whether Th17 cells could serve as a novel immunotherapy target to improve the outcomes of leukemia treatment remains challenging. Furthermore, the increased presence of Th17 cells in the leukemia niche and the relationship between IL-17A and leukemia progression in vivo requires further investigation. Here, we found that in the B-ALL microenvironment, IL-17A secreted by Th17 cells promoted the proliferation and survival of B-ALL cells and increased the secretion of the chemokine CXCL16 by leukemia cells. This, in turn, further promoted the recruitment of Th17 cells to the leukemia microenvironment. However, importantly, Th17 cells have a high degree of plasticity and can transdifferentiate into other lineage subsets depending on the microenvironment[50]. It remains unclear whether Th17 cells are capable of transdifferentiating into other lineage subsets, such as Th1 or Treg cells, in the B-ALL microenvironment.

We found that IL-17A was highly expressed in Ph⁺ B-ALL patients and that high expression of IL-17A was associated with poor clinical outcomes. The established BCR-ABL-induced B-ALL mouse model provides an excellent tool for investigating the related molecular mechanism and validating the therapeutic efficacy. IL-17A deficiency or anti-IL-17A treatment significantly inhibited leukemia cell proliferation activity and increased the survival rates of B-ALL mice. More strikingly, B-ALL mice treated with both anti-IL-17A and the TKI imatinib achieved complete remission. Monoclonal antibodies targeting IL-17A, including secukinumab and ixekizumab, have been approved to treat moderate to severe plaque psoriasis[13,14]. Although some researchers are cautious about the dual role of Th17/IL-17A signaling in tumor progression[11], numerous studies have demonstrated that IL-17A can enhance angiogenesis, proliferation, metastasis or drug resistance in multiple cancers, which supports the potential value of blocking IL-17A as an antitumor therapy[15]. Our study reveals that anti-IL-17A antibodies may achieve significant therapeutic efficacy in Ph⁺ B-ALL patients under appropriate conditions, especially in combination with TKI therapy. Moreover, Salvestrini et al. reported that the leukemia cells homing to the bone marrow niche resist chemotherapeutic drugs[51]. In this study, we found that IL-17A promoted but anti-IL-17A monoclonal antibodies (mAbs) treatment reduced leukemia cells' homing, suggesting that anti-IL-17A mAbs treatment may render leukemia cells more sensitive to standard chemotherapeutic drugs.

It has been reported that IL-17A treatment induces the activation of the MAPK/ERK pathway through the phosphorylation of c-RAF, P42/P44 MAPK, MEK and ERK in human nasal endothelial cells[52], HUVECs[53], macrophages[54], and breast cancer cells[55]. Additionally, Mazzera et al. reported that activated MEK1/2 could assemble into a pentameric complex with BCR-ABL1, BCR, and ABL1, leading to the phosphorylation of BCR and BCR-ABL1 at Tyr360 and Tyr177[56]. We also determined the effect of IL-17A treatment on the phosphorylation of MEK. The phosphorylation of MEK was increased in B-ALL cells following IL-17A treatment. Thus, IL-17A may increase the phosphorylation of BCR-ABL1

through MEK activation. Bi et al. also reported that IL-17A promoted the proliferation of Ph⁻ B-ALL cells through activation of the PI3K/AKT and Jak2/Stat3 signalling pathways[16]. Here, we found that IL-17A treatment promoted the proliferation of Ph⁺ B-ALL cells by activating the IL6/JAK/STAT3 signalling pathway, which is consistent with the previous study. Additionally, we found that IL-17A treatment increased the activation of the BCR-ABL signalling pathway. The BCR-ABL oncoprotein is a key molecular basis for leukemia pathogenesis, playing important roles in cell proliferation, survival, and immunosuppression through the activation of several downstream signalling pathways, such as the JAK/STAT, PI3K/AKT, and Raf/MEK/ERK pathways[57]. Thus, targeting the IL-17A signalling pathway could be a valuable therapeutic approach for B-ALL.

NF-кB is an important regulator involved in the gene transcription of chemokines[58,59]. In this study, we showed that IL-17A could induce secretion of the chemokine CXCL16 in the leukemia niche by promoting the nuclear translocation and subsequent activity of NF-кB. In the nucleus, NF-кB binds to the CXCL16 promoter region and increases CXCL16 transcription. The mechanism by which IL-17A was found to induce CXCL16 secretion in B-ALL cells in this study is similar to previous findings indicating that LPS or oxidative stress generated by $H_2O_2$ induced CXCL16 expression via the NF-кB signalling pathway[60,61]. CXCL16 is characterized as a proangiogenic cytokine and acts as an important angiogenic factor in the tumor microenvironment[62]. Suppression of CXCR6, a receptor for CXCL16, has been found to reduce tumor angiogenesis in a hepatocellular carcinoma xenograft mouse model[63]. Moreover, targeting CXCL16 either in cancer cells using shRNA or in the microenvironment using a neutralizing antibody efficiently blocks tumor growth and angiogenesis in thyroid cancers[64]. In our study, we verified the roles of CXCL16 in Ph⁺ B-ALL in vivo. We found that the depletion of CXCL16 with an anti-CXCL16 neutralizing antibody attenuated leukemia progression and decreased the frequency of Th17 cells in the leukemia niche in vivo, suggesting that IL-17A mediates leukemia progression by promoting CXCL16 secretion. Therefore, CXCL16 might be another therapeutic target and be suitable for use in a precision medicine approach for Ph⁺ B-ALL treatment, especially in patients with elevated CXCL16 expression.

Here, we found that Th17 cells are enriched in the niche microenvironment. The cytokine IL-17A, derived from Th17 cells, promotes Ph⁺ B-ALL progression and increases CXCL16 expression in leukemia cells. The increased amount of CXCL16 further enhances Th17 cell differentiation and recruitment, creating a positive loop that drives Ph⁺ B-ALL progression in the BM niche. Importantly, we further confirmed that targeting the Th17 niche or deleting CXCL16 inhibits Ph⁺ B-ALL progression and exhibits a synergistic therapeutic effect in combination with imatinib in vivo. Our study provides valuable insights for the development of a therapeutic approach combining anti-IL-17A or anti-CXCL16 neutralizing antibodies with the TKI imatinib to achieve complete remission of Ph⁺ B-ALL.

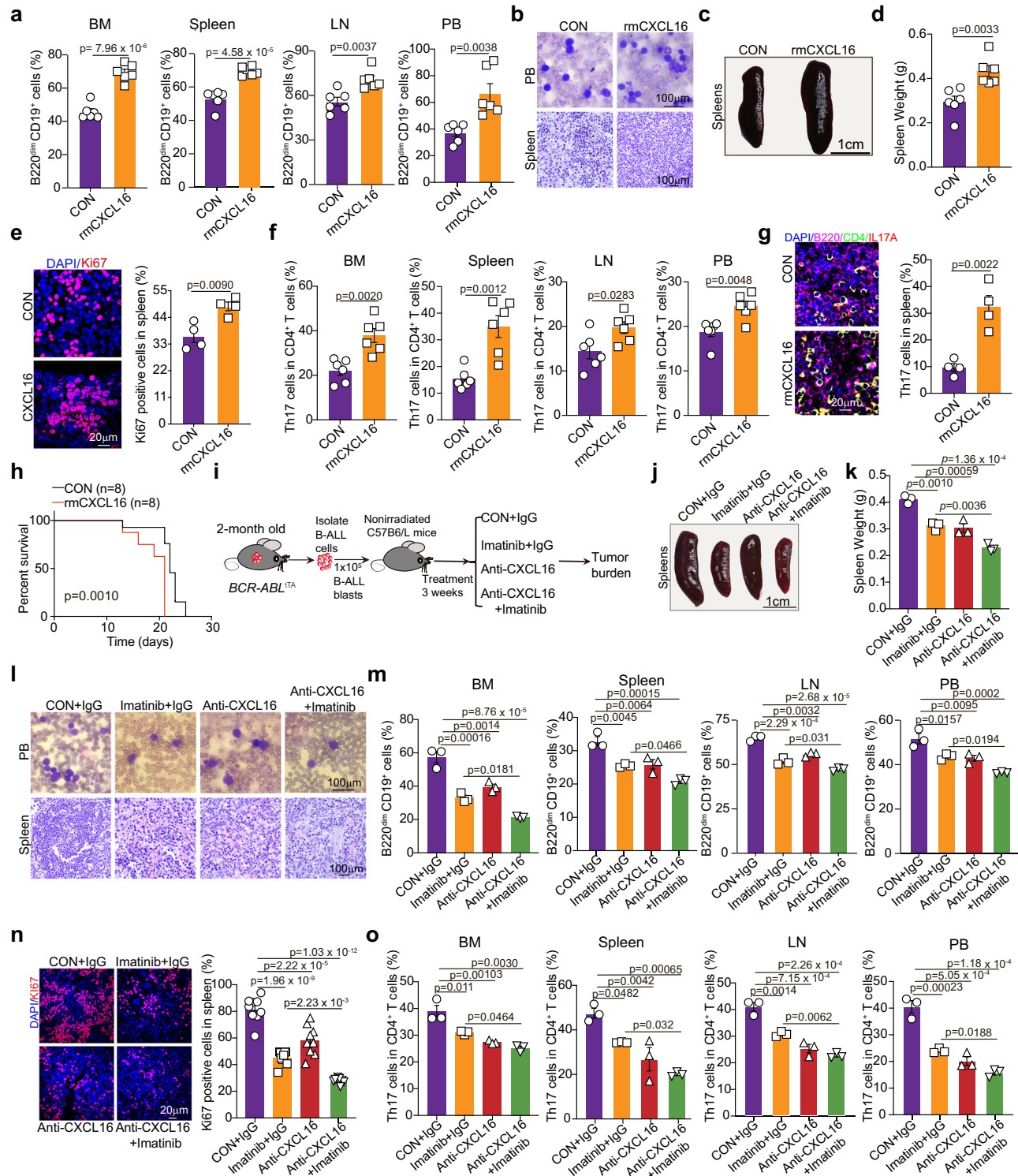

## Methods

### Study approval

Human B-ALL patient specimens were obtained from the Institute of Hematology and Blood Diseases Hospital of Peking Union Medical College (PUMC). Informed consent was obtained from all participants in accordance with the Declaration of Helsinki, and the informed consent of participants under the age of 18 was obtained from their parents or guardians. The procedure was approved by the institutional review board at the Ethics Committee of the Institute of Hematology and Blood Diseases Hospital of PUMC (IIT2021011-EC-1). Our study is compliant with the 'Guidance of the Ministry of Science and Technology (MOST) for the Review and Approval of Human Genetic Resources', which requires formal approval for the export of human genetic material or data from China. The patient-related information are available in Supplementary Data 1.

### Animal studies

NOD-SCID IL2Rg-null (NSG) mice (6-8 weeks old, male) were purchased from the Nanjing Biomedical Research Institute of Nanjing University (Nanjing, China). The tetO-BCR/ABL1 (B6.FVB/N-Tg(tetO-BCR/ABL1)

**Fig. 7 | The anti-CXCL16 mAb synergizes with imatinib to attenuate the progression of Ph⁺ B-ALL. a** Flow cytometric analysis of the percentages of B220^dim CD19⁺ cells in BM, spleens, LNs and PB of mice with secondary transplantation treated with or without rmCXCL16 ($n = 6$ mice per group). **b–d** Representative images of Wright-Giemsa-stained PB smears (**b**, top), H&E staining of spleens (**b**, bottom), spleens (**c**) and spleen weights (**d**) from the indicated mice ($n = 6$ mice per group). **e** Representative images of Ki67 staining in the spleen tissues from the indicated mice. $n = 4$ fields, two different mice per group. **f** Flow cytometric analysis of the percentages of Th17 cells in the PB, LNs, spleens and BM from the indicated mice ($n = 6$ mice per group). **g** Spleen tissues from the indicated mice were subjected to immunofluorescence staining for IL-17A (red), CD4 (green), and B220 (rose red). Representative images of Th17 cells were shown. $n = 4$ fields, two different mice per group. **h** Kaplan–Meier survival curves for the indicated mice ($n = 8$ mice per group). **i** Schematic strategy for investigating the effects of anti-CXCL16 mAb alone or combined with imatinib on Ph⁺ B-ALL progression. **j–l** Representative

images of spleens (**j**), spleen weights (**k**), Wright-Giemsa-stained PB smears (**l, top**), and H&E staining of the spleen (**l, bottom**) from leukemia mice treated with the indicated agents ($n = 3$ mice per group). **m** Flow cytometric analysis of the percentages of B220^dim CD19⁺ cells in the PB, BM, LNs and spleens from leukemia mice treated with the indicated agents ($n = 3$ mice per group). **n** The percentage of Ki-67⁺ cells in the spleen was detected by immunofluorescence staining in the indicated mice. Data are presented as means ± S.E.M of eight random fields of view from three different mice per group. **o** Flow cytometric analysis of the percentages of Th17 cells in the PBMCs, LNs, spleens and BM of leukemia mice treated with the indicated agents ($n = 3$ mice per group). (**a**, **m**) The gating strategy for B220^dim CD19⁺ cells was shown in Supplementary Fig. 3a. (**f**, **o**) The gating strategy for Th17 cells in the CD4⁺ T cells was shown in Supplementary Fig. 6b. Statistical significance was calculated by (**a**, **d**–**g**) two-tailed Student's t-test; (**k**, **m**–**o**) one-way ANOVA with Tukey's multiple comparison tests; Data are presented as means ± S.E.M. Source data are provided as a Source Data file.

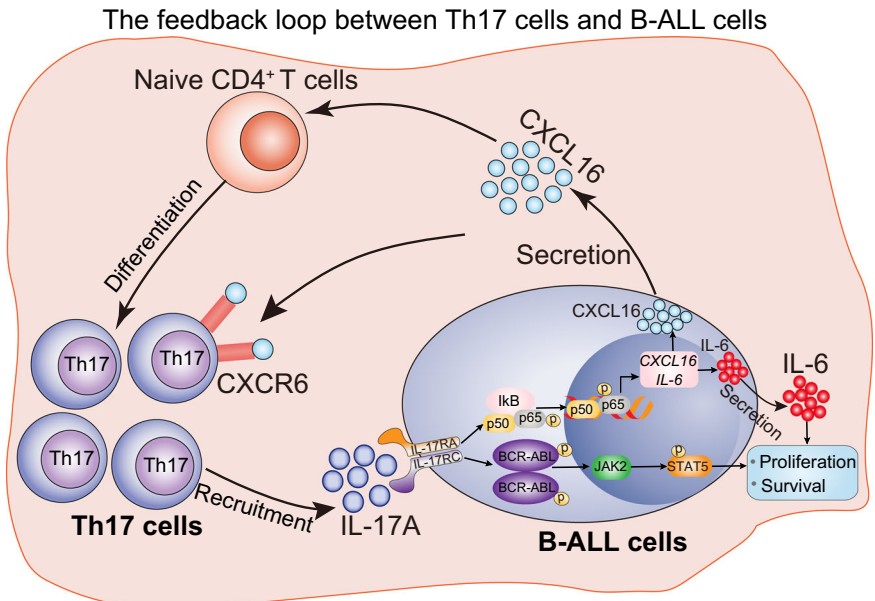

**Fig. 8 | Schematic diagram illustrating the feedback loop formed by Th17 cells, IL-17A, and CXCL16 that promotes Ph⁺ B-ALL progression.** The Th17 cell population and IL-17A expression are distinctively increased in Ph⁺ B-ALL patients, and high expression of IL-17A promotes the progression of Ph⁺ B-ALL. IL-17A promotes the proliferation and survival of Ph⁺ B-ALL cells by activating the BCR-ABL

and IL6/JAK/STAT3 signaling pathways. Moreover, IL-17A can increase the secretion of the chemokine CXCL16 from leukemia cells by activating NF-kB, which in turn mediates the differentiation and recruitment of Th17 cells to the leukemia niche microenvironment. Targeting IL-17A or CXCL16 in the leukemia niche microenvironment attenuates the progression of Ph⁺ B-ALL.

2Dgt/Nju) mice (5–6 weeks old, 1 male and 2 females, strain #N00005) were purchased from the Nanjing Biomedical Research Institute of Nanjing University (Nanjing, China). These mice had been backcrossed over 10 generations onto the C57BL/6 background. MMTV-tTA (B6.Cg-Tg(MMTVtTA)1Mam/J) mice (6–8 weeks old, 2 males and 2 females, Strain #002618) were purchased from The Jackson Laboratory (CA, USA). C57BL/6 J mice (female, 6-8 weeks old) were purchased from Hua Fu Kang Technology Co., Ltd. (Beijing, China). *BCR-ABL^tTA* mice were generated by crossing female tetO-BCR/ABL1 (B6.FVB/N-Tg(tetO-BCR/ABL1)2Dgt/Nju) mice with male MMTV-tTA mice (B6.Cg-Tg(MMTVtTA)1Mam/J) under continuous administration of tetracycline (0.5 g/L) in the drinking water. Withdrawal of tetracycline in *BCR-ABL^tTA* mice resulted in the development of Ph⁺ B-ALL within 1.5-2 months[35]. *IL-17A^−/−* (C57BL/6Smoc-*Il17a*^em1Smoc) mice (5–6 weeks old, 1 male and 2 females, Cat. NO. NM-KO-00131) were obtained from Shanghai Model Organisms Center, Inc.. To determine the role of IL-17A in leukemogenesis, founder lines of *BCR-ABL^tTA* mice that had features of human B-ALL and scored positive for Ph⁺ B-ALL were used. A total of $1 \times 10^5$ mouse B-ALL-like spleen cells from *BCR-ABL^tTA* mice were intravenously injected into nonirradiated *IL-17A^−/−* mice or *WT* mice at 8–10 weeks of

age. All mice were maintained in the animal facility at the Institute of Materia Medica under specific-pathogen-free (SPF) conditions. Animals were monitored daily. If mice manifested symptoms such as failure to thrive, weight loss > 10% of total body weight, hunching, activity decrease (stationary unless stimulated or hind limb paralysis), bleeding, infection, and/or fatigue, they were euthanized immediately, as per protocol approved by the Animal Experimentation Ethics Committee of the Chinese Academy of Medical Sciences. For animal studies, the mice were earmarked before grouping and then randomly separated into groups by an independent person; however, no particular randomization method was used. The sample size was predetermined empirically according to previous experience using the same strains and treatments. Generally, we used $N \geq 3$ mice per genotype and condition. We ensured that the experimental groups were balanced in terms of animal age and weight. All animal procedures were conducted according to the guidelines of the Institutional Committee for the Ethics of Animal Care and Treatment of the Chinese Academy of Medical Sciences (CAMS) and Peking Union Medical College (PUMC). All animal procedures were consistent with the ARRIVE guidelines.

## Primary leukemia cell purification/isolation

B cells from mouse spleens, lymph nodes (LNs) or bone marrow (BM) were isolated with Dynabeads Mouse Pan B (B220) (Dynabeads; Invitrogen, 11441D). Human primary B-ALL cells were purified using MACS CD19 MicroBeads (Miltenyi Biotec, 130-050-301), and the percentage of B-ALL cells (CD19+) was determined to be > 90% by flow cytometry.

## Mouse serum cytokine array

Cytokine expression was measured in the serum (10-fold dilution) of *WT* mice (n = 3) and *BCR-ABL*tTA mice using a Quantibody® Mouse TH17 Cytokines Array (RayBiotech, QAM-TH17-1) and Quantibody® Mouse Chemokine Array (RayBiotech, QAM-CHE-1). Data were collected using an InnoScan 300 Microarray Scanner instrument following the manufacturer's instructions and analyzed by Q-Analyzer Tool.

## Mouse model of secondary transplantation of *BCR-ABL*tTA B-ALL cells

All recipient mice used for transplantation, including *WT* and *IL-17A* KO mice, were on a C57BL/6 background. The original tetO-BCR/ABL1 mice were generated on the FVB background[35]. We obtained tetO-BCR/ABL1 mice from Nanjing Biomedical Research Institute of Nanjing University; these mice had been backcrossed over 10 generations onto the C57BL/6 background. These tetO-BCR/ABL1 mice were then crossed with MMTV-tTA transactivator (B6.Cg-Tg(MMTVtTA)1Mam/J) mice on the C57BL/6 background to generate double transgenic (*BCR-ABL*tTA) mice. Since the donor *BCR-ABL*tTA mice and recipient mice were congenic on the same background, no additional backcrossing was performed before transplantation. After 1.5-2 months of tetracycline administration, the first-generation *BCR-ABL*tTA mice (G1) were sacrificed. For primary transplantations, $5 \times 10^6$ BM or spleen cells were intravenously injected into nonirradiated female C57B6/L mice between 6 and 8 weeks old. For secondary transplantations, mouse B-ALL-like spleen cells from primary transplanted mice were intravenously injected into female nonirradiated C57B6/L mice ($1 \times 10^5$ cells per mouse) between 6 and 8 weeks old. Mice transplanted with leukemia cells were randomly assigned to either treatment. Three days after transplantation, 70 mg/kg imatinib (Sigma–Aldrich, SML1027, p.o., once a day), 5 mg/kg anti-IL-17A neutralizing antibody (Bio X Cell, BP0173, i.v., twice a week), or 0.5 mg/kg anti-mouse CXCL16 neutralizing antibody (R&D, MAB503, i.v., twice a week) were administered for 3 consecutive weeks. The mice in the control group were treated with IgG. For CXCL16 administration, rmCXCL16 protein (PeproTech, 250-28) in 100 μL of PBS (50 μg/kg) was administered by intraperitoneal injection every 3 days for 2 weeks. The mice in the control group were treated with an identical volume of PBS. The mice were sacrificed by excessive anesthesia 14 days after cytokine administration. To evaluate the survival rate, these mice were monitored for 50 days.

## Murine Xenograft Model of Leukemia

To evaluate the effect of IL-17A on the progression of Ph+ B-ALL in vivo, NOD-SCID IL2Rg-null (NSG) mice (6-8 weeks old, male) were intravenously injected with $2 \times 10^6$ GFP-luc tagged SupB15 cells. One-half hour later, in vivo animal imaging was performed to monitor the development of B-ALL in NOD-SCID mice using the IVIS Spectrum optical imaging system (PerkinElmer Inc., OH, USA) at different time points (day 1, day 8, and day 15). Then, beginning 4 days after SupB15 cell transplantation, 50 μg/kg rhIL-17A was administered for 3 weeks (twice a week), and the survival rate of these mice was monitored for 50 days.

## Bioluminescence imaging and analysis

A total of $2 \times 10^6$ GFP-luc tagged SupB15 cells were intravenously injected into NSG mice (6-8 weeks old, male). The infiltration of B-ALL cells was monitored on days 1, 8, and 15 after transplantation with a bioluminescence imaging system. For bioluminescence imaging, mice were anesthetized and injected with 1.5 mg of D-luciferin (i.p., 15 mg/ml in PBS). Imaging was completed between 2 and 5 min after injection and was performed with an IVIS SpectrumCT In Vivo Imaging System coupled to live image acquisition and analyzed with the Living Image Software (version 3.2) (Perkin Elmer, OH, USA). For plotting bioluminescence imaging data, the photon flux was calculated for each mouse by using a rectangular region of interest encompassing the thorax of the mouse in a prone position.

## Flow cytometry

Surface antigens of primary cell suspensions from the BM, PBMCs, spleen, or LNs were stained with fluorophore-conjugated antibodies in FACS buffer for 30 min at room temperature (RT) in the dark. For intracellular staining, cells were stimulated with 30 nM PMA (Sigma–Aldrich, P1585) and 1 μM ionomycin (Sigma–Aldrich, I3909) in the presence of 2 μg/ml brefeldin A (Selleck, S7046) for 4–6 h at 37 °C. Following surface staining, the cells were fixed with fixation buffer (BD, 554655), and intracellular staining was performed using 1x permeabilization buffer (BD, 554723) for 30 min at RT. The gating strategies for mouse CD4+ T cells and mouse Th17 cells in BM, PB, LN, and Spleen are shown in Supplementary Fig. 1i and Supplementary Fig. 6b. The following fluorescently labeled antibodies against surface proteins were used for human/mouse cell staining: mouse anti-CXCL16 antibody (R&D, AF503, 1:100), followed by staining with secondary antibodies labeled with Alexa Fluor 555; PE-conjugated anti-mouse/human CD45R/B220 antibody (BioLegend, 103208, 1:100); APC/Cyanine7-conjugated anti-mouse CD19 antibody (BioLegend, 152412, 1:100); PerCP-conjugated anti-mouse CD19 antibody (BioLegend, 115531, 1:100), Alexa Fluor 647-conjugated anti-mouse Ly-51 (BP-1) antibody (BioLegend, 108311, 1:100), PE/Cyanine7-conjugated anti-mouse CD43 antibody (BioLegend, 143209, 1:100), APC-conjugated anti-mouse CD4 antibody (BioLegend, 116014, 1:100), FITC-conjugated anti-mouse IL-17A antibody (BioLegend, 506907, 1:100), PE/Cyanine7-conjugated anti-mouse Ki-67 antibody (BioLegend, 652426, 1:100), PerCP anti-mouse CD19 Antibody (BioLegend, 115531, 1:100), PE-conjugated anti-human CD19 antibody (BioLegend, 302254, 1:100), FITC-conjugated anti-human CD4 antibody (BioLegend, 357405, 1:100), APC-conjugated anti-human CD4 antibody (BioLegend, 317416, 1:100), FITC-conjugated anti-mouse/human Ki-67 antibody (BioLegend, 151212, 1:100), PE/Cyanine7-conjugated anti-human IL-17A antibody (BioLegend, 512315, 1:100), Alexa Fluor 647-conjugated anti-human IL-17A antibody (BioLegend, 512310, 1:100), APC-conjugated human IL-17RA/IL-17R antibody (R & D, FAB177A, 1:100), and Alexa Fluor® 488-conjugated human IL-17RC antibody (R & D, FAB22691,1:100). The gating strategies for B220dim CD19+ and B220dim CD19+ Ki-67+ cells in the BM, PB, LN, and Spleen are shown in Supplementary Fig. 3a, b. To evaluate Th17 cell frequencies, bone marrow mononuclear cells (BMMCs) isolated from Ph+ B-ALL patients and HDs were stimulated with 30 nM PMA (Sigma–Aldrich, P1585) and 1 μM ionomycin (Sigma–Aldrich, I3909) in the presence of 2 μg/ml brefeldin A (Selleck, S7046) for 4-6 hr at 37 °C. After incubation, cell surface staining with an APC-conjugated anti-human CD4 antibody (BioLegend, 317416, 1:100) was performed at room temperature in the dark for 30 min. The cells were subsequently fixed, permeabilized and stained with a PE/Cyanine7-conjugated anti-human IL-17A antibody (BioLegend, 512315, 1:100). The gating strategies for human CD4+ T cells in BM and human Th17 cells in the CD4+ T cells are shown in Supplementary Fig. 1a. For analysis of the proliferation of human leukemia cells, CD19+ leukemia cells were loaded with 2.5 μM CFSE (eBioscience, 65-0850-84, 1:1000) according to the manufacturer's instructions. FCS EXPRESS or FlowJo 10.8.1 software was used for data analysis.

## Cell culture

The acute B-cell leukemia cell lines SupB15, BV173, and NALM-6 were purchased from Cell Resource Center, Peking Union Medical College,

where they were recently authenticated by short tandem repeat (STR) profiling and characterized by mycoplasma detection. The cells were cultured and maintained in RPMI 1640 medium supplemented with 10% fetal bovine serum (Invitrogen, CA, USA) under 5% carbon dioxide. SupB15 cells stably expressing GFP-Luc were achieved by infecting cells with GFP-Luc lentivirus particles, and stable GFP-Luc-overexpressing SupB15 cells were selected by sorting for GFP$^+$ cells by flow cytometry. Stable Su-B15-GFP-Luc cell lines were cultured in RPMI 1640 medium containing 5 μg/ml puromycin (Gibco, CA, USA). All cell lines were verified negative for mycoplasma contamination by MycoAlert™ Mycoplasma Detection Kit (Lonza, LT07-318).

## Luciferase reporter assay

To test whether IL-17A regulates the transcriptional activity of NF-kB, the NF-kB reporter gene plasmid and luciferase reporter constructs were stably transfected into HEK-293 cells, which were then treated with different concentrations of recombinant human IL-17A (Pepro-Tech, 200-17). After 24 h, luciferase activity was detected with a luminometer using a Luciferase Reporter Assay System (Promega, Madison, WI, E1910) in accordance with the manufacturer's instructions.

## Homing assay

A total of $1 \times 10^7$ GFP-luc-tagged SupB15 cells treated with or without rhIL-17A (50 μg/kg per mouse) were transplanted into NSG mice. A total of $1 \times 10^7$ primary mouse B-ALL cells were labelled with CFSE (eBioscience, 65-0850-84, 1:1000), treated with or without 5 mg/kg anti-IL-17A and intravenously injected into 6- to 8-week-old C57BL/6 mice. For homing assays in *IL-17A* mice, $1 \times 10^7$ primary mouse B-ALL cells labelled with CFSE (eBioscience, 65-0850-84, 1:1000) were intravenously injected into 6- to 8-week-old *WT* mice or *IL-17A$^{-/-}$* mice. The frequency of GFP$^+$ cells or CFSE$^+$ cells was determined in the BM and spleen 16 h after transplantation by flow cytometric analysis.

## RNA-seq library preparation and sequencing

RNA purity was assessed using a Kaiao K5500® Spectrophotometer (Kaiao, Beijing, China). RNA integrity and concentration were assessed using the RNA Nano 6000 Assay Kit of the Bioanalyzer 2100 system (Agilent Technologies, CA, USA). The RNA integrity number (RIN) was > 7.5 for all samples. Sequencing libraries were generated using an NEBNext® Ultra™ RNA Library Prep Kit for Illumina® (#E7530L, NEB, USA) following the manufacturer's recommendations, and index codes were added to attribute sequences to each sample. In brief, mRNA was purified from 2 μg of each total RNA sample using poly-T oligo-attached magnetic beads. Using divalent cations under elevated temperature, fragmentation was performed in NEB Next First Strand Synthesis Reaction Buffer (5X). First-strand cDNA was synthesized using random hexamer primers and RNase H. Second-strand cDNA synthesis was subsequently performed using buffer, dNTPs, DNA polymerase I and RNase H. The library fragments were purified with QiaQuick PCR kits and eluted with EB buffer, and then A-tailing and adapter additions were implemented for terminal repair. The intended products were retrieved, and PCR was performed to complete the library preparation.

## Gene Set Enrichment Analysis (GSEA)

Publicly available microarray data of ALL patients were retrieved from the Gene Expression Omnibus (GEO) (GSE13204). Of these cases, the upper tenth (58 patients, IL-17A positively correlated cases) had the highest levels of IL-17A expression. In contrast, the lower tenth (58 patients, IL-17A negatively correlated) had the lowest levels of IL-17A expression. Preranked GSEA was performed using a gene set of the BCR-ABL signaling-associated genes. For RNA-seq data, GSEA was performed using the clusterProfiler package by R 4.0.3.

## Real-time PCR

Total RNA was extracted from cell samples using TRIzol (Invitrogen, CA, USA) following the manufacturer's instructions. According to the manufacturer's instructions, reverse transcription of the total cellular RNA was performed using a TransScript One-Step gDNA Removal Kit (TransGen Biotech, AE311) and a cDNA synthesis superMix Kit (Trans-Gen Biotech, AE341). PCR amplification was performed in triplicate. Each reaction contained 1X SYBR FAST qPCR Master Mix (KAPA BIO-SYSTEMS, KK4602), 1 μL of mixed primers and 1 μL of template cDNA. PCR was conducted with a MyCycler thermal cycling instrument (qTOWER, Analytik Jena, German). The PCR primer sequences are listed as follows: human *IL-13* forward: 5′-ACGGTCATTGCTCTCAC TTGCC-3′, human *IL-13* reverse: 5′-CTGTCAGGTTGATGCTCCATACC-3′; human *CCL20* forward: 5′- AAGTTGTCTGTGTGCGCAAATCC -3′, human *CCL20* reverse: 5′- CCATTCCAGAAAAGCCACAGTTTT -3′; human *CCL-19* forward: 5′- CGTGAGGAACTTCCACTACCTTC-3′, human *CCL-19* reverse: 5′- GTCTCTGGATGATGCGTTCTACC-3′; human *IL-5* forward: 5′- GGAATAGGCACACTGGAGAGTC -3′, human *IL-5* reverse: 5′- CTCTCCGT CTTTCTTCTCCACAC -3′; human *CXCL16* forward: 5′- CCTATGTGCTGTGCAAGA GGAG -3′, human *CXCL16* reverse: 5′- CTGGGCAACATAGAGTCCGTCT -3′; human *CCL-5* forward: 5′- CCT GCTGCTTTGCCTACATTGC -3′, human *CCL-5* reverse: 5′- ACACACTTG GCGGTTCTTTCGG -3′; human *CCL-24* forward: 5′- TGAGAACCGA GTGGTCAGCTAC -3′, human *CCL-24* reverse: 5′- TTCTGCTTGGCGT CCAGGTTCT -3′; human *CCL-1* forward: 5′- ACCAGCTCCATCTGCTC CAATG -3′, human *CCL-1* reverse: 5′- TGTGCCTCTGAACCCATCCAAC -3′; human *CCL-17* forward: 5′- TTCTCTGCA GCACATCCACGCA -3′, human *CCL-17* reverse: 5′- CTGGAGCAGTCCTCAGATGTCT -3′; human *CXCL5* forward: 5′- CAGACCACGCAAGGAGTTCATC -3′, human *CXCL5* reverse: 5′- CAGACCACGCAAGGAGTTCATC -3′; human *CCL-22* forward: 5′- TCCTGGGTTCAAGCGATTCTCC -3′, human *CCL-22* reverse: 5′- GTCAGGAGTTCAAGACCAGCCT -3′; human *CXCL-11* forward: 5′- AAGG ACAACGATGCCTAAATCCC -3′, human *CXCL-11* reverse: 5′- CAGAT GCCCTTTTCCAGGACTTC -3′; human *CXCL-2* forward: 5′- GGCAGAAA GCTTGTCTCAACCC -3′, human *CXCL-2* reverse: 5′- CTCCTTCAGGAAC AGCCACCAA -3′; human *CCL-15* forward: 5′- TGATGTCAAAGCTTCCA CTGGAAA -3′, human *CCL-15* reverse: 5′- GAGTGAACACGGG ATGCTTTGTG-3′; human *BCR-ABL* forward: 5′- TCCACTCAGCCACTGG ATTTAA -3′, human *BCR-ABL* reverse: 5′- TGAGGCTCAAAGTCAGAT GCTACT -3′; human *IL-6* forward: 5′- AGACAGCCACTCACCTCTTCAG -3′, human *IL-6* reverse: 5′- TTCTGCCAGTGCCTC TTTGCTG -3′; human *GAPDH* forward, 5′-GTGGACATCCGCAAAGACC-3′, human *GAPDH* reverse, 5′-CCTAGAAGCATTTGCGGTG-3′; mouse *P65* forward: 5′- GCT GCCAAAGAAGGACACGACA -3′, mouse *P65* reverse: 5′- GGCAGGCTA TTGCTCATCACAG -3′; mouse *Jak2* forward: 5′- GCTACCAGATGGA AACTGTGCG -3′, mouse *Jak2* reverse: 5′- GCCTCTGTAATGTTGGTGAG ATC -3′; mouse *Cxcl16* forward: 5′- GCAGGGTACTTTGGATCACATCC -3′, mouse *Cxcl16* reverse: 5′- AGTTCACGGACCCACTGGTCTT -3′; mouse *Il-6* forward: 5′- TACCACTTCACAAGTCGGAGGC -3′, mouse *Il-6* reverse: 5′- CTGCAAGTGCATCATCGTTGTTC -3′; and mouse *Gapdh* forward: 5′- CATCACTGCCACCCAGAAGACTG -3′, mouse *Gapdh* reverse: 5′- ATGCCAGTGAGCTTCCCGTTCAG-3′.

## ELISA

Plasma samples were collected from peripheral blood (PB) after centrifugation and stored at −80 °C until use. The IL-17A, CXCL16, IL-6, IL-5, CXCL5, or CCL5 concentrations in human or mouse plasma or in cell culture supernatant were measured by using commercial ELISA kits according to the manufacturer's instructions. These kits included the Human IL-17A ELISA Kit (Biotrend, CHE0054), Human CXCL16 ELISA Kit (Biotrend, CHE01117), Human IL-6 ELISA Kit (Biotrend, CHE0009), Mouse CXCL16 ELISA Kit (mlbio, ml002119), Human IL-5 ELISA Kit (Biotrend, CHE0006), Human CXCL5 ELISA Kit (Biotrend, CHE0162), and Human Rantes ELISA Kit (Biotrend, CHE0092).

## Immunoblotting and immunofluorescence staining

Proteins were extracted from cultured cells with RIPA lysis buffer (Beyotime Technology). The protein concentrations were measured using BCA protein assay kits. The protein extracts were separated by SDS-polyacrylamide gel electrophoresis (SDS–PAGE) and subjected to immunoblot analysis. Images were obtained with a Tanon 5200 chemiluminescence imaging system (Tanon, Shanghai). For immunoblotting, B-ALL cells were treated with IL-17A (200 ng/ml) and prepared with Nuclear and Cytoplasmic Extraction Reagents (Beyotime, P0028). Equal amounts of protein extracts were electrophoresed on 10% SDS–PAGE gels and transferred onto PVDF membranes. The anti-NF-kappaB p65 (D14E12) antibody (CST) #8242, 1:1000), anti-phospho-NF-kappaB p65 (93H1) antibody (CST, #3033 T, 1:1000), anti-BCR antibody (CST) 3902 S, 1:1000), anti-phospho-Bcr (Tyr177) antibody (CST, #3901, 1:1000), anti-Stat5 (D2O6Y) antibody (CST, #94205, 1:1000), anti-phospho-Stat5 (Tyr694) (D47E7) antibody ((CST, #4322, 1:1000), anti-Stat3 (D1B2J) antibody (CST, #30835, 1:1000), anti-phospho-Stat3 (Tyr705) (D3A7) antibody (CST, #9145, 1:1000), anti-AKT(pan) (11E7) antibody (CST, #4685, 1:1000), anti-phospho-Akt (Ser473) (D9E) antibody (CST, #4060, 1:1000), anti-MEK1/2 (D1A5) antibody (CST, #8727, 1:1000), anti-phospho- MEK1/2 (Ser217/221) (41G9) (CST, #8727, 1:1000), anti-Histone H3 (D1H2) XP (CST, # 60932, 1:1000), anti-Lamin A/C antibody (CST, #4777, 1:1000), anti-β-Actin (8H10D10) antibody (CST, #3700, 1:1000), and anti-GAPDH (OTI2D9) antibody (ZSGB-Bio, TA-08, 1:5000) were used as primary antibodies. The indicated protein band intensities were quantified using ImageJ software and normalized to those of the bands representing the loading controls (for total protein and cytoplasmic proteins, GAPDH was used as the loading control; for nuclear proteins, Lamin A/C was used as the loading control). For immunofluorescence staining, sections were fixed with 4% paraformaldehyde, permeabilized with 0.5% Triton X-100 at room temperature (RT) for 15 min, and blocked with Superblock™ (Thermo Scientific™ 37515) at RT for 1 h. Subsequently, 50 μl droplets containing an anti-mouse B220/CD45R antibody (R&D, MAB1217, 1:100), anti-CD4 antibody (Abcam, ab183685, 1:100), anti-IL-17A antibody (Abcam, ab189377, 1:100) or anti-CXCL16 antibody (R&D, AF503, 1:100) were placed on Parafilm in a dark humidified chamber. The samples were incubated at 4 °C overnight. After washing 3 times, Alexa Fluor 488 (Thermo Fisher, R37114, 1:200)-, Alexa Fluor 555 (Thermo Fisher, A31572, 1:200)-, Alexa Fluor 647 (Thermo Fisher, A21247, 1:200)-, Alexa Fluor 647 (Thermo Fisher, A21447, 1:200)- or Alexa Fluor 555 (Thermo Fisher, A21434, 1:200)-conjugated goat anti-rat IgG (H + L) secondary antibodies were added, and incubation was performed as described in the previous section for primary antibody staining. The coverslips were then incubated for 1 hr at RT. The nuclei were stained with DAPI. Images were captured with an Olympus confocal microscope (Olympus Microsystems, Fv3000, CA, USA). Quantitative image analysis was performed with Imaris 9.3.1 software.

## Coculture of B-ALL cells with Th17 cells

Cell suspension was isolated from BM or PB of Ph+ B-ALL patients. Then, the cell suspension was lysed by a red blood cell lysis buffer to remove red cells and passed through a 40 μm cell strainer. Th17 cells were sorted based on staining with a FITC-conjugated anti-human CD4 antibody (BioLegend, 357405, 1:100) and a PE-conjugated anti-human CD196 (CCR6) antibody (BioLegend, 399003, 1:100) by a FACS Aria III cell sorter (BD Bioscience). The sorting strategy for Th17 cells is shown in Supplementary Fig. 6a. B-ALL cells were isolated from patients and sorted by using MAC CD19 MicroBeads (Miltenyi Biotec, 130-050-301). For the coculture assay, viable Th17 cells were cocultured with B-ALL cells at a ratio of 1:1, and recombinant human CXCL16 (PeproTech, 300-55), recombinant human IL-17A (PeproTech, 200-17), the anti-human IL-17A neutralizing antibody (Bio X Cell, SIM0013), the anti-human CXCL16 neutralizing antibody (R&D, AF976), anti-goat IgG (R&D, AB-108-C) or recombinant human IgG1 Fc (Bio X Cell, BE0096)

was added to the indicated coculture systems. The proliferation activity and apoptosis of leukemia cells were detected based on staining with the PE-conjugated anti-human CD19 antibody (BioLegend, 302254, 1:100), the APC-conjugated anti-human CD4 antibody (BioLegend, 317416, 1:100), the FITC-conjugated anti-human CD4 antibody (BioLegend, 344604, 1:100), the FITC-conjugated anti-mouse/human Ki-67 antibody (BioLegend, 151212, 1:100), or Annexin V/PI (Dojindo, AD11). For migration assays, Th17 cells were placed in the upper chamber containing a 5 μm pore polycarbonate membrane insert (Corning, 3421), and B-ALL cells were cultured in the lower chamber. After 12 h, the cells in the lower chamber were counted with a cell counter, and the percentage of CD4+ T cells in the lower chamber was determined by FCS analysis. The migrated Th17 cells in the lower chambers of the inserts were quantified as follows: cell number (lower chamber) x percentage of CD4+ T cells in the lower chamber.

## Th17 cell in vitro differentiation

Naïve CD4+ T cells were isolated from the spleens of C57 mice using the MagniSort Mouse Naïve CD4+ T-Cell Enrichment Kit (Miltenyi, 130-104-453) according to the manufacturer's instructions. A total of $1 \times 10^6$ naïve CD4+ T cells were cultured in 1 ml of RPMI-1640 complete medium containing 2 μg/ml anti-CD3 antibody (Biogems, 05112-25), 2 μg/ml anti-CD28 antibody (Biogems, 10312-25), 20 ng/ml recombinant murine IL-6 (PeproTech, 216-16), 4 ng/ml recombinant human TGF-β (PeproTech, 100-21), 50 μg/ml anti-IL-4 antibody (Biogems, 81112-25), or 50 μg/ml anti-mouse IFN-γ antibody (Biogems, 80812-25) with or without different concentrations of rmCXCL16 for 6 days. For Th17 cell in vitro differentiation, cells were stimulated with 30 nM PMA and 1 μM ionomycin in the presence of 2 μg/ml brefeldin A for 4–6 h at 37 °C. The cells were subsequently stained with an APC-conjugated anti-CD4 antibody, fixed, permeabilized and then intracellularly stained with a FITC-conjugated anti-IL-17A antibody. The stained cells were acquired and analyzed using FCS express software.

## DFS and OS analysis

The data was generated from published data available on data set GSE11877. The B-ALL samples were divided into two groups based on the best separation cutoff value of IL-17A expression (6.64, 35% of IL-17A expression). "IL-17A high" refers to the group of B-ALL patients with *IL-17A* mRNA expression above the cutoff value of 6.64, while "IL-17A low" refers to the group of B-ALL patients with *IL-17A* mRNA expression below the cutoff value of 6.64. DFS (disease-free survival) and OS (overall survival) were analyzed by the Kaplan–Meier method.

## Statistical analysis

The data are expressed as the mean ± standard error of the mean (SEM). Comparisons between the two groups were performed by unpaired Student's t-test with Tukey's multiple comparison tests. Comparisons between multiple groups were performed by one-way ANOVA. Correlations between groups were determined by Pearson correlation analysis. The survival rates were analyzed by the Kaplan–Meier method. The sample number (n) indicates the number of independent biological samples in each experiment. The sample numbers and experimental repeats are shown in the figures and figure legends. Generally, all experiments were carried out with $n \geq 3$ biological replicates. A value of $p < 0.05$ was considered to indicate statistical significance. The analyses were performed using GraphPad Prism 9.2.0 software.

## Reporting summary

Further information on research design is available in the Nature Portfolio Reporting Summary linked to this article.

# Data availability

The RNA-seq data generated in this study have been deposited in the NCBI Gene Expression Omnibus (GEO) database under accession code

GSE210091. The single-cell datasets from HDs and Ph+ B-ALL patients were analyzed using the accession code GSE134759. The overall survival (OS) and disease-free survival (DFS) of patients with high *IL-17A* and in low *IL-17A* expression were analyzed in data set GSE11877. IL-*17 RA* and *IL-17 RC* expression in Ph⁺ B-ALL patients, Ph⁻ B-ALL patients and Healthy Donors and enrichment of gene sets in Ph⁺ B-ALL patients with *IL-17 RA* high vs *IL-17 RA* low or *IL-17 A* high vs *IL-17 A* low were analyzed using the accession code GSE13204. The remaining data are available within the Article, Supplementary Information or Source Data file. Source data are provided in this paper. Source data are provided with this paper.

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

## Acknowledgements

This work was supported by grants from the National Key R&D Program of China (2022YFA1106100), the National Natural Science Foundation of China (82173853 and 82373914 to B.C., 82222070 and 82273973 to K.L., 82003798 to F.W., 82173379 to J.J.Y., 82073892 to J.M.Y.), the CAMS Innovation Fund for Medical Sciences (2021-I2M-1-026 to B.C.; 2021-I2M-1-030 to K.L.; 2021-I2M-1-070 to T.-t.Z.), the CAMS Innovation Engineering Platform Fund for Medical Sciences (2022-I2M-2-002), the Fundamental Research Funds for the Central Universities (2022-RC350-07, 3332022149), Chinese Academy of Medical Sciences (CAMS) Central Public-interest Scientific Institution Basal Research Fund (2018PT35004) and Beijing Outstanding Young Scientist Program (BJJWZYJH01201910023028).

## Author contributions

B.C., K.L., and E.L.J. conceptualized the study and participated in the overall design, supervision and coordination. F.W., Y.X.L., Z.N.Y., and W.B.C. designed and performed most of the experiments. Y.L., L.Y.Z., T.T.Z., and C.X.Z. participated in the molecular and cellular biological experiments. J.M.Y., J.J.Y., and J.C. Z performed animal studies. X.W.Z., P.P.L., M.Z.H., B.W.L.N., and S.Z.F. contributed to scientific discussion and data interpretation. K.L., B.C., F.W., and Y.X.L. wrote the manuscript. Z.W.H (Deceased April 26th, 2021) participated the initial design of the study from 2019 to 2021. All authors have read and approved the manuscript.

## Competing interests

The authors declare no competing interests.
