## [Peer Review File · Nature Communications]

Targeting IL-17A enhances imatinib efficacy in Philadelphia chromosome-positive B-cell acute lymphoblastic leukemiaEditorial Note: Parts of this Peer Review File have been redacted as indicated to remove third-party material where no permission to publish could be obtained.

REVIEWERS' COMMENTS:

Reviewer #1 (Remarks to the Author): with expertise in B-ALL, bone marrow niche

In this manuscript by Wang et al. the authors suggest that T helper type 17 (Th17) cells and IL-17A are increased in BCR-ABL1+ B-ALL, conferring an adverse prognosis. IL-17A activates the IL6/JAK/STAT3 pathway in B-ALL cells, while also activating NF-KappaB, which leads to increased secretion of CXCL16. CXCL16 in turn supposedly increases Th17 differentiation, thereby perpetuating this circuit. The authors show that deficiency or inhibition of IL-17A or CXCL16 potentiates the anti-leukemic effects of imatinib. Overall, the manuscript is of interest and it tackles the important aspect of the immune microenvironment in leukemia. However, there are several concerns, which diminish enthusiasm substantially:

1. Importantly, the origin and applicability of the murine model is unclear and of doubtful relevance. What is the reference for this model? If this is the BCR/ABL Tsr/J model, as specified by the authors, this model reportedly does not give rise to leukemia (Foley et al., Cell Reports, 2013). Other BCR-ABL transgenic models give rise to chronic myeloid leukemia, but not B-ALL. This needs to be clarified. Peripheral blood results, blood smears of good quality in less thick areas and flow cytometry results need to be shown.
2. Why is IL-17A specific for BCR-ABL1+ B-ALL? What are the levels in non-BCR-ABL1+ B-ALL?
3. Figure 1F: Why is mRNA expression of IL17A lower and protein expression higher in BCR-ABL1+ B-ALL?
4. Figure 1G: Please provide IL-17RA and -RC expression compared to healthy cells.
5. Do leukemia cells secrete IL-17A?
6. For how many generations were all recipient mice backcrossed prior to transplantation? Insufficient backcrossing can interfere with transplant results.
7. The authors start their various treatments very early after transplantation, i.e. on day 2. It is possible that homing and certainly not engraftment were completed at that time. If treatment is initiated too early, the homing and engraftment of transplanted leukemia cells will be impaired. Homing assays (before 18 hours) need to be performed for all treatment

experiments, i.e. IL-17A, anti-IL17A etc.

8. Please show homing in IL-17A KO mice.
9. How does IL-17A lead to increased phosphorylation of BCR-ABL1?
10. What do Th17 cells do in the B-ALL microenvironment? Please discuss.
11. How did the authors identify CXCL5, IL-5, CXCL16 and CCL5?
12. Discussion, line 330: Where do the authors show that CXCL16 promotes Th17 differentiation?
13. Why do the authors use brefeldin A for intracellular staining?
14. Figure 1A/B: Please show CD4+ cells also.
15. Figure 1D: There seem to be huge differences between mice. More mice are needed.
16. Please show that Th17 cells make IL-17A.
17. Figure 2L and 3N: Why are the survival curves of the control mice so different (40 days versus 25 days)?
18. How did the authors arrive at the concentration of IL-17A (100 ng/ml)?
19. Figure 5A: IL-17A staining seems absent. Why?
20. Figure 5C: Please also use a BCR-ABL1-negative cell line.
21. Figure 6C: For how long were cells treated?
22. Do Th17 cells express CXCR6??

Minor:

1. The authors repeatedly speak of mouse models (e.g. p. 6 line 111), but at this point in the manuscript only one model has been tested.
2. The manuscript needs editing for English syntax and grammar.
3. More information is needed in the legends.
4. The authors repeatedly misuse the term 'xenotransplantation'. This only refers to transplantation between species.
5. The discussion is largely a summary of the findings and needs to be improved.
6. Figure 2A: Schematic: Lymphocytes=Lymphoblasts
7. Quantifications of Western Blots are needed, as differences seem slight.

Reviewer #2 (Remarks to the Author): with expertise in Th17/IL17 and hematological

malignancies

Wang et al reported that the Th17 cells were enriched in niche microenvironment that promoted Ph+ B-ALL progression and increased CXCL16 expression in leukemia cells. They found that Th17 cells and Ph+ B-ALL cells promoted each other through IL-17A and CXCL16, forming a positive loop to enhance Ph+ B-ALL progression in the bone marrow niche. They further validated IL-17A or CXCL16 as a therapeutic target for treatment of Ph+ B-ALL alone or in combination with imatinib. The study was well designed and executed and the results are compelling. Overall finding is significant because the study provides the rationale and means to target IL-17A or CXCL16 for controlling B-ALL. However, there are several comments and questions for the authors to consider for further strengthening this excellent work. Specific comments are:

- 1) Th17/IL-17 has been reported before by others to promote B cell lymphoma (e.g. Ferretti E et al, 2015; Bi L et al, 2016). The authors should cite the previously published studies and highlight the novelty of the current study.
- 2) How Th17 cells were detected and defined should be clearly stated.
- 3) IL-17A was used at 100ng/ml or even higher in the experiments. The concentrations are clearly beyond physiological or even pathological levels, which questions the relevance of the observed results. What is the range of IL-17A levels in vivo in mice or patients with B-ALL? Can IL-17A induce any signal sub nanogram levels?
- 4) Th17 cells also produce IL-21 and IL-22. Was either cytokine increased in B-ALL? If so, what is their contribution to B-ALL progress?

Reviewer #3 (Remarks to the Author): with expertise in Th17/IL17 and hematological malignancies

In this manuscript by Wang et al, the authors systemically described a feed-back loop between Ph+ ALL and Th17 cells through the CXCL16/IL-17A axis which is accountable for adverse prognosis of this disease. Mechanistically, IL-17A promoted the progression of Ph+ ALL through activation of BCR-ABL, STAT3 and NF- κ B pathways whereas inhibition of IL-17A attenuated the disease progression. The authors further demonstrated a feed-back loop between IL-17A and CXCL16 predominantly in vitro. More interestingly, inhibition of either

IL-17A or CXCL16 was synergistic with imatinib for the treatment of Ph+ ALL. Although the correlation of IL-17A with disease progression in acute and chronic leukemia have been previously reported (in particular, Ref #22), the present study conducted deeper mechanistical investigations. Overall, this is a very interesting study and the findings are clinically translatable. However there are a few points that need further investigation or clarification.

Major comments:

- 1.The present study focuses on the interaction between Th17 cells (though IL-17A) and Ph+ ALL (CXCL16) which represents a direct crosstalk. There is no evidence or data that support a contribution from other factors in the microenvironment. Removal of “leukemia microenvironment” from the title can make the message of the story clearer.
- 2.The y-axis should be shown from 0 in all figures (for example 2C, 5G-J, 6B, 6J, 7A, 7E). The truncated y-axis exaggerates the difference and sometimes is misleading. For example, the difference between groups in Fig 6B/F are minimal (< 5 – 10%) but looks significantly bigger on the plots. As such, it’s not convincing that the small changes in CXCL16 is accountable for the Th17 differentiation. In addition, the increase of CXCL16 in Ph+ ALL (Figure 6C) may reflect higher leukemia burden rather than an increase of expression on the per cell basis. What’s the difference of systemic CXCL16 in the mouse model (IgG vs anti-IL-17A, WT vs BCR/ABL-tTA)?
- 3.Please clarify the rationale to adopt xenograft model in Figure 7A-G. It’s also very confusing that rmCXCL16 is effective in this model since it doesn’t have target in NSG mice. Please also describe the xenograft model in the methods section. What’s the effect of rmCXCL16 treatment in BCR/ABL-tTA model which has the target?
- 4.It’s interesting to know if T cells (by depletion of CD4 or CD4/CD8 T cells) are beneficial or detrimental for the control of Ph+ ALL since T cells are significantly inhibited by chemotherapy and can be better targeted.

Minor comments

- 5.Ref #22 demonstrated that IL-17A promotes B-ALL proliferation by activating Akt/STAT3 (through IL-17RA). This should be cited and discussed in more detail.
- 6.The BCR/ABL-tTA mouse model should be described briefly in the results (or introduction) section.
- 7.Has CXCL16 been tested in the chemokine assay (Figure 1D)?

8. TNF alpha should be written as TNF since TNF beta has been renamed.

Reviewer #4 (Remarks to the Author): with expertise in B-ALL, tumor microenvironment

The authors describe a role for Th17 cells in promoting inflammation in the bone marrow of Ph+ B-ALL patients, and suggest blockade of IL-17 or of Th17 recruitment as a novel therapeutic strategy in B-ALL. While the idea of inflammation as a driver of B-ALL progression is interesting and important for the field, the article has several technical issues that have to be addressed to allow for correct interpretation of the role of Th17 cells in B-ALL.

1. In figure 1 (panels A-B), the authors claim that Th17 cells are increased in Ph+ B-ALL patients and mouse models, however it is unclear how the cells were detected (what gating strategy was used to identify Th17 cells?). Also, there are several single cell datasets from B-ALL published out there. Can the authors use them to further validate their Th17 findings?
2. In the cytokine assay (Fig 1D), the authors show an averaged column for all 3 WT mice and 3 separate columns for the 3 different BCR-ABL mice. It would be better to show separated data for the 3 WT mice as well. Also, showing z-scores would be more meaningful than showing averages, especially since it seems like all WT mice were normalized to 1.
3. In the Kaplan Meier curves in Fig 1H, it's not clear what the cutoff for high/low IL17A expression is – how were the groups determined?
4. In Figure 2, the authors demonstrate the effect of IL-17 on B-ALL cells. In panel E, the quantification of cells is unclear and it would be better to show bar plots with cell percentages for each condition/time. In 2I, the authors claim an effect for IL-17 in promoting B-ALL progression in vivo, here it would be good to show imaging data at an earlier time point, before IL-17 treatment started, to demonstrate that disease burden in the mice was equal before treatment. In the vehicle group, there are no signs of disease in 5 out of 8 mice at the time point shown, which could suggest a technical issue with engraftment. Similarly, the tissue infiltration and survival data (Fig 2K-L) could also be affected by this.
5. In figure 4, the authors describe activation of inflammatory pathways in B-ALL cells following IL-17A exposure. In Fig. 4A, it would be good to highlight on the volcano plot some

of the genes upregulated or downregulated following IL-17A treatment.

6. The authors treat B-ALL cells with IL-17A and demonstrate activation of inflammatory cytokine transcription and of the JAK/STAT pathway. Here the authors use a high concentration of IL-17A (100ng/mL), whereas in the ELISA shown in figure 1 the concentrations measured in B-ALL patients were much lower (10-20 pg/mL). Would the same effect be seen when using physiological IL-17 concentrations? similarly in figure 6 the authors use very high concentrations of IL-17A. In addition, the western blot for pSTAT3 and p-p65 in panel 4F is hard to interpret – the authors should quantify the levels of pSTAT3 and p-p65 and perhaps use a shorter exposure for p-p65.

7. In figure 5 the authors describe the crosstalk between B-ALL cells and Th17 cells. the migration assay shown in panel B is unclear – how was migration measured? Was the effect specific for Th17 cells, or were similar effects seen with other T cell subsets (CD4+/CD8+)? The authors use a B-ALL cell line, did the T cells used in the assay match the HLA of the cell line?

8. In figure 5C the authors demonstrate an increase in transcription of several chemokines following IL-17A treatment – can the authors demonstrate secretion of these by ELISA?

9. The authors perform co-culture of T cells derived from healthy donors with primary B-ALL samples – were the HLA isotypes matched for these assays? Can the authors perform the same experiment with T cells derived from the same patient as the tumor cells?

Reviewers' comments:

Reviewer #1 (Remarks to the Author): with expertise in B-ALL, bone marrow niche In this manuscript by Wang et al. the authors suggest that T helper type 17 (Th17) cells and IL-17A are increased in BCR-ABL1+ B-ALL, conferring an adverse prognosis. IL-17A activates the IL6/JAK/STAT3 pathway in B-ALL cells, while also activating NF-KappaB, which leads to increased secretion of CXCL16. CXCL16 in turn supposedly increases Th17 differentiation, thereby perpetuating this circuit. The authors show that deficiency or inhibition of IL-17A or CXCL16 potentiates the anti-leukemic effects of imatinib. Overall, the manuscript is of interest and it tackles the important aspect of the immune microenvironment in leukemia. However, there are several concerns, which diminish enthusiasm substantially:

1. Importantly, the origin and applicability of the murine model is unclear and of doubtful relevance. What is the reference for this model? If this is the BCR/ABL Tsr/J model, as specified by the authors, this model reportedly does not give rise to leukemia (Foley et al., Cell Reports, 2013). Other BCR-ABL transgenic models give rise to chronic myeloid leukemia, but not B-ALL. This needs to be clarified. Peripheral blood results, blood smears of good quality in less thick areas and flow cytometry results need to be shown.

Response: Thanks for your astute observation and professional suggestion. We apologize for the writing mistakes in the name of *BCR-ABL* mice with “BCR/ABL Tsr/J” in the original manuscript (line 401). In fact, the mice used in this study are “tetO-BCR/ABL1” mice, which were developed in a study by Huettner et al. (Huettner et al., 2000), not “BCR/ABL Tsr/J” mice. We moved forward to change the FVB background with the C57BL/6 background with over ten generations of backcrossing. We apologize for this mistake, which caused concern about the appropriateness of the mouse model.

Here, we also provide detailed information on B6.FVB/N-Tg(tetO-BCR/ABL1)2Dgt/Nju mice and purchasing records in the following figures. In the revised manuscript (page 22, lines 413-420), we have corrected the name of the *BCR-ABL* mouse strain, provided additional information about the origin and applicability of the murine model, and cited the relevant reference as the following.

【 Methods section 】 tetO-BCR/ABL1 (B6.FVB/N-Tg(tetO-BCR/ABL1)2Dgt/Nju) mice (strain #N00005) were purchased from the Nanjing Biomedical Research Institute of Nanjing University

(Nanjing, China). MMTV-tTA (B6.Cg-Tg(MMTVtTA)1Mam/J) mice (Strain #002618) were purchased from The Jackson Laboratory. C57BL/6J mice (8-10 weeks old) were purchased from Hua Fu Kang Technology Co., Ltd. (Beijing, China). *BCR-ABL^{tTA}* mice were generated by crossing female tetO-BCR/ABL1 (B6.FVB/N-Tg(tetO-BCR/ABL1)2Dgt/Nju) mice with male MMTV-tTA mice under continuous administration of tetracycline (0.5 g/L) in the drinking water. Withdrawal of tetracycline in *BCR-ABL^{tTA}* mice resulted in the development of Ph+ B-ALL within 1.5-2 months (Huettner et al., 2000).

Furthermore, we have provided detailed information about the *BCR-ABL^{tTA}* mice in the Results section (page 6, lines 94-106). Withdrawal of tetracycline administration in double transgenic mice (*BCR-ABL^{tTA}*; C57BL/6 (C57) background) induced the expression of BCR-ABL1 and resulted in the development of B-cell leukemia in 100% of the mice (Below panel, Supplementary Fig 1c). The longest survival time of *BCR-ABL^{tTA}* mice after tetracycline withdrawal was about 15 weeks (Below panel, Supplementary Fig 1d). Necropsy demonstrated massive splenomegaly and enlargement of the majority of lymph nodes (LNs) in these mice (Below panel, Supplementary Fig 1e-1f). FACS analysis showed that the splenic lymphoblasts from the diseased mice were B220^{dim} (Below panel, Supplementary Fig 1g, left) and expressed CD19, CD43, and BP-1 (Below panel, Supplementary Fig 1g, right). This pattern suggested that the transformed cells underwent arrest at the pre-B-cell stage of development and was similar to the cell surface marker expression pattern identified in Ph+ B-ALL patient samples and MMTV-*BCR-ABL^{tTA}* transgenic mice (FVB background) (Harb et al., 2008; Hardy et al., 1991). Additionally, the frequency of immature blasts was found to be increased in Wright-Giemsa-stained peripheral blood (PB) and HE-stained BM and spleen sections of *BCR-ABL^{tTA}* mice compared to wild-type (WT) mice (Below panel, Supplementary Fig 1h).

Fig. Ph+ B-ALL development in tetO-BCR/ABL1 mice with C57BL/6 background.

[Editorial Note: Redacted]

Fig. Genotyping reports and purchase records for the B6.FVB/N-Tg(tetO-BCR/ABL1)2Dgt/Nju mice

2. Why is IL-17A specific for BCR-ABL1+ B-ALL? What are the levels in non-BCR-ABL1+ B-ALL?

Response: Thank you for your professional suggestion. BCR-ABL1+ (Ph⁺) B-ALL represents the most common subtype of adult ALL (20%-30%) and accounts for about 50% of all cases in older adults. Older patients with Ph⁺ ALL have poor survival due to the inability to tolerate intensive chemotherapy and ineligibility for myeloablative allogeneic HCT (Mathisen et al., 2011). Therefore, we initially focused on exploring the role of elevated IL-17A in Ph⁺ B-ALL in this study. With your suggestion, we measured the IL-17A concentration in the serum of healthy donors (HDs), newly diagnosed Ph⁺ B-ALL patients, and Ph⁻ B-ALL patients. The IL-17A concentration was increased in both Ph⁻ B-ALL and Ph⁺ B-ALL patients compared to healthy donors. The concentration of IL-17A was higher in Ph⁺ B-ALL patients than in Ph⁻ B-ALL patients (Below panel, Fig 1h). Notably, it was reported that IL-17A also promoted the proliferation of Nalm-6 cells (Ph⁻ B-ALL cell line)(Bi et al., 2016). This observation suggests that IL-17A may also affect Ph⁻ B-ALL cells, not specifically Ph⁺ B-ALL cells. We added these comments in the discussion section.

Fig 1h

3. Figure 1F: Why is mRNA expression of IL17A lower and protein expression higher in BCR-ABL1+ B-ALL?

Response: Thanks for your questions. We noticed that protein expression of IL-17A is higher in BCR-ABL1^{tTA} mice and Ph⁺ B-ALL patients. We then investigated the mRNA and protein expression for the receptor of IL-17A. In the original Fig 1F (Revised Fig 1h), we first examined the expression of IL-17RA and IL-17RC in Ph⁺ B-ALL patients in the GEO database (GSE13204). We observed that both IL-17RA and IL-17RC were expressed in Ph⁺ B-ALL patients. However, the mRNA level of IL-17RA

in Ph⁺ B-ALL patients was even lower than that in healthy donors. This led us to study the protein expression of IL-17 RA and IL-17 RC because proteins perform essential biological functions. We then further investigated the protein levels of IL-17 RA and IL-17RC in primary Ph⁺ B-ALL cells by flow cytometric analysis. These results confirmed the positive surface expression of IL-17 RA and IL-17 RC on Ph⁺ B-ALL cells (Original **Figure 1G**, also **Revised Fig 1i**).

4. *Figure 1G: Please provide IL-17RA and -RC expression compared to healthy cells.*

Response: Following your suggestion, we measured the expression of IL-17RA and IL-17RC in healthy B cells and Ph⁺ B-ALL cells. Both IL-17RA and IL-17RC could be detected in these cells by flow cytometry (**Below panel, Fig 1i**). Interestingly, the expression of IL-17RC was significantly higher in Ph⁺ B-ALL cells than in normal B cells, whereas the expression of IL-17RA remained similar (**Below panel, Fig 1i**).

5. *Do leukemia cells secrete IL-17A?*

Response: IL-17A is primarily produced by Th17 cells. Recent studies indicated that IL-17A can be made by other cell types, such as macrophages, dendritic cells, $\gamma\delta$ T cells, NK and NKT cells, CD8⁺ T cells, neutrophils, and tumor cells (Onishi and Gaffen, 2010). However, it remains unclear whether B-ALL cells secrete IL-17A. In this study, we first measured the IL-17A expression in the supernatants of SupB15 cells with or without IL-23 plus lipopolysaccharide (LPS) stimulation by ELISA. Surprisingly, we did not detect the IL-17A in the supernatants of SupB15 cells cultured under either standard conditions or IL-23 plus LPS stimulation. To further confirm these findings, we measured IL-17A expression in primary mouse B-ALL cells by flow cytometry. Consistent with the ELISA results, we observed minimal secretion of IL-17A by B-ALL cells under standard culture conditions, and the expression of IL-17A did not increase by IL-23 plus LPS stimulation (**Below panel, panel A**). Collectively, our data suggest that B-ALL cells secrete almost no IL-17A. These data are shown below and will not be incorporated in the revised manuscript.

Fig IL-17A expression by flow cytometry. (A, left) Representative FACS plots and quantification of the percentage of IL-17A-positive B-ALL cells among mouse B-ALL cells with or without IL-23 (10 ng/mL) plus LPS (1 μ g/mL) stimulation for 24 h or 48 h. The percentage of IL-17A-positive B-ALL cells was determined by flow cytometry. **(A, right)** The data are presented as the mean \pm S.E.M. of 3 independent experiments. Statistical significance was determined by a two-tailed Student's t-test.

6. For how many generations were all recipient mice backcrossed prior to transplantation? Insufficient backcrossing can interfere with transplant results.

Response: Thank you for your professional suggestion. All recipient mice, including *WT* and IL-17A-knockout (KO) mice, were on the C57BL/6 background. The original tetO-BCR/ABL1 mice were generated on the FVB background (Huettner et al., 2000). We obtained tetO-BCR/ABL1 mice from Nanjing Biomedical Research Institute of Nanjing University; these mice had been backcrossed over ten generations onto the C57BL/6 background. These tetO-BCR/ABL1 mice were then crossed with MMTV-tTA transactivator (B6.Cg-Tg(MMTVtTA)1Mam/J) mice on the C57BL/6 background to generate double transgenic (*BCR-ABL^{tTA}*) mice. Since the donor *BCR-ABL^{tTA}* mice and recipient mice were congenic on the same background, no additional backcrossing was performed before transplantation. We have added a detailed description of the background of the *BCR-ABL^{tTA}* mice to the Methods section of the revised manuscript (Page 24, lines 456-462).

7. The authors start their various treatments very early after transplantation, i.e. on day 2. It is possible that homing and certainly not engraftment were completed at that time. If treatment is initiated too early, the homing and engraftment of transplanted leukemia cells will be impaired. Homing assays (before 18 hours) need to be performed for all treatment experiments, i.e. IL-17A, anti-IL17A etc.

Response: Thanks for your suggestion. It is actual and possible that homing and engraftment were not completed in the early phase after transplantation. In this study, we initiated IL-17A treatment on day 2, while the other treatments including anti-IL17A antibodies, anti-CXCL16 antibodies, or CXCL16, were started three days after transplantation. We described the treatment initiation timing in the Methods section of the original manuscript (lines 454-455) as follows: "Mice implanted with leukemic cells were randomly assigned to either different type of treatment. Three days after transplantation,

70 mg/kg imatinib...". In the original study, we believed the homing of transplanted leukemia cells was completed three days after transplantation.

Following your guide, we first performed homing assays (prior to 18 hours) to investigate the effects of IL-17A treatment on the homing of transplanted leukemia cells (GFP-luc-tagged SupB15 cells). Flow cytometric analysis showed that IL-17A treatment enhanced leukemia cells' homing to the bone marrow (BM) and spleen (Below panel, Supplementary Fig 2b-2c). Additionally, we used CFSE-labeled leukemia cells isolated from *BCR-ABL*^{t7A} mice to determine the effect of anti-IL-17A treatment on leukemia cell homing. The results indicated that anti-IL-17A treatment reduced the homing of leukemia cells to the BM and spleen in recipient mice (Below panel, Supplementary Fig 3b).

To clarify the effect of rhIL-17A treatment on the engraftment of B-ALL cells, we then performed rhIL-17A treatment four days after SupB15 cells transplantation (Below panel, Fig 2g). This treatment strategy should avoid impairing the homing and engraftment of transplanted leukemia cells. The engraftment results showed that rhIL-17A treatment promoted Ph⁺ B-ALL progression (Below panel, Fig 2h and Supplementary 2d), as indicated by the increased infiltration of leukemia cells in the peripheral blood, BM, and spleen (Below panel, Fig 2i-2j) and decreased survival rates of leukemia-engrafted mice (Below panel, Fig 2k).

Supplementary Fig 2b**Fig 2g****Supplementary Fig 2c****Supplementary Fig 3b****Fig 2h****Supplementary Fig 2d****Fig 2i****Fig 2j****Fig 2k**
8. Please show homing in IL-17A KO mice.

Response: Following your suggestion, we performed homing assays in IL-17A knockout mice using CFSE-labeled leukemia cells isolated from *BCR-ABL*^{TA} mice. We examined the distribution of leukemia cells in the BM and spleen of *WT* and IL-17A-knockout (*IL-17A*^{-/-}) mice 16 hours after transplantation. Flow cytometric analysis showed that *IL-17A* knockout reduced the homing of leukemia cells to the BM and spleen (Below panel, Supplementary Fig 3a).

Supplementary Fig 3a
9. How does IL-17A lead to increased phosphorylation of BCR-ABL1?

Response: It was reported that IL-17A induced the activation of the MAPK/ERK pathway through the phosphorylation of c-RAF, P42/P44 MAPK, MEK, and ERK in human nasal endothelial cells (Li et al.,

2022a), HUVECs (Xing et al., 2013), macrophages (Chen et al., 2013), and breast cancer cells (Mombelli et al., 2015). Moreover, Mazzera et al. reported that activated MEK1/2 could assemble into a pentameric complex with BCR-ABL1, BCR, and ABL1, leading to the phosphorylation of BCR and BCR-ABL1 at Tyr360 and Tyr177 (Mazzera et al., 2023). Here, we also determined the effect of IL-17A treatment on the phosphorylation of MEK. The activity of MEK was enhanced in B-ALL cells following IL-17A treatment (**Below panel, Fig 4i**). Thus, IL-17A may increase the phosphorylation of BCR-ABL1 through MEK activation, and this assumption has been incorporated into the Discussion section of the current revised manuscript (**page 19, lines 367-374**).

Fig 4i

10. *What do Th17 cells do in the B-ALL microenvironment? Please discuss.*

Response: Thank you for your professional suggestion. Th17 cells accumulate specifically in various tumors, indicating the occurrence of targeted recruitment of these cells by the tumor microenvironment. Th17 cells play a crucial role in inflammation and tumor immunity by secreting Th17-associated cytokines such as IL-17A, IL-21, IL-23, and TNF (Najafi and Mirshafiey, 2019). Our study found that in the B-ALL microenvironment, IL-17A secreted by Th17 cells promoted the proliferation and survival of B-ALL cells and enhanced the secretion of the chemokine CXCL16 by leukemia cells. This, in turn, further promoted the recruitment of Th17 cells into the leukemia microenvironment. However, Th17 cells have a high degree of plasticity and can transdifferentiate into other lineage subsets depending on the microenvironment (Guery and Hugues, 2015). Whether Th17 cells can transdifferentiate into different lineage subsets, such as Th1 or Treg cells, in the B-ALL microenvironment remains unclear. We have added these descriptions to the Discussion section of the revised manuscript (**page 17, lines 334-336 and lines 345-351**).

11. *How did the authors identify CXCL5, IL-5, CXCL16 and CCL5?*

Re: To identify the chemokines involved in migrating Th17 cells to leukemia cells, we first utilized qPCR to examine the impact of IL-17A on chemokine expression in leukemia cells. We observed significant increases in the mRNA levels of *CXCL5*, *IL-5*, *CXCL16*, and *CCL5* in both SupB15 and BV173 cells following the rhIL-17A treatment. Based on these findings, we then selected *CXCL5*, *IL-5*, *CXCL16* and *CCL5* for further investigation. The presentation of the qPCR results in the original

Figure 5C was unclear, leading to confusion. In the revised manuscript, we have reordered the presentation of the genes in descending order of fold change and included a corrected version of Figure 5c (Below panel, Fig 5c).

12. Discussion, line 330: Where do the authors show that CXCL16 promotes Th17 differentiation?

Response: We showed these data in the original Figure 5G. Naïve CD4⁺ T cells were isolated and cultured in Th17 differentiation medium with or without CXCL16. Flow cytometry indicated that CXCL16 significantly triggered the differentiation of Th17 cells in a dose-dependent manner (Below panel, Revised Fig 5h).

13. Why do the authors use brefeldin A for intracellular staining?

Response: To enhance intracellular staining and prevent the secretion of cytokines and other secreted proteins, brefeldin A is commonly utilized because it inhibits protein transport. We referred to the studies conducted by Salazar et al. and Pesce et al. as references for staining Th17 cells (Pesce et al., 2022; Salazar et al., 2020).

14. Figure 1A/B: Please show CD4⁺ cells also.

Response: Following your suggestion, we have shown the percentages of CD4⁺ T cells in bone marrow mononuclear cells (BMMCs) from human Ph⁺ B-ALL patients and PB from *BCR-ABL*^{TA} mice in the revised manuscript. Although the frequency of CD4⁺ T cells among BMMCs was slightly decreased in patients with Ph⁺ B-ALL compared to healthy donors, there was no statistically significant difference between these two groups (Below panel, Supplementary Fig 1b). We also

analyzed the percentage of CD4⁺ T cells in the PB of mice with BCR-ABL-driven B-ALL (*BCR-ABL^{tTA}* mice) throughout B-ALL progression. The proportion of CD4⁺ T cells in the PB of mice with BCR-ABL-driven B-ALL was similar to that of *WT* mice during B-ALL progression (**Below panel, Supplementary Fig 1i**).

Supplementary Fig 1b **Supplementary Fig 1i**

15. Figure 1D: There seem to be huge differences between mice. More mice are needed.

Response: Following your suggestion, we measured the plasma concentrations of Th1/Th2/Th17 cytokines and chemokines in a larger sample size of *BCR-ABL^{tTA}* B-ALL mice and *WT* mice. These results showed significant increases in Fractalkine, MIP-1a, CXCL16, BLC, and IL-17A, among others, in *BCR-ABL^{tTA}* B-ALL mice compared to *WT* mice. Notably, IL-17A was among the top 5 of the 42 examined cytokines (**Below panel, Fig 1f**).

Fig 1f

16. Please show that Th17 cells make IL-17A.

Response: Following your suggestion, we have presented the flow cytometry plot and gating strategy of Th17 cells, including IL-17A staining, in Supplementary Fig 1a.

Supplementary Fig 1a

17. Figure 2L and 3N: Why are the survival curves of the control mice so different (40 days versus 25 days)?

Response: Thank you for your astute observation. The discrepancy in survival curves between control mice can be attributed to using different mouse models. Specifically, in original Figure 2L (Revised Figure 2k), we used a NOD-SCID xenograft model (human Ph⁺ B-ALL cell transplantation

model) to evaluate the impact of IL-17A on Ph⁺ B-ALL progression *in vivo*. However, in the original Figure 3N (**Revised Fig 3n**), we utilized a mouse model of syngeneic transplantation of *BCR-ABL*^{IT^A} B-ALL cells. Thus, the dissimilarity in the survival curves shown in original Figure 2L (**Revised Fig 2k**) and original Figure 3N (**Revised Fig 3n**) can be attributed to the different mouse models employed.

18. How did the authors arrive at the concentration of IL-17A (100 ng/ml)?

Response: The selected concentration of IL-17A (100 ng/ml) was initially based on published papers in which 50 ng/ml or 100 ng/ml IL-17A was used to stimulate leukemia cells (Bi et al., 2016; Han et al., 2014; Li et al., 2015). This helps us to reach the platform or maximum effects of IL-17A in the system, which might mimic the niche microenvironments. Thanks for your suggestion. We evaluated the impact of different IL-17A concentrations (0, 25 ng/ml, 50 ng/ml, 100 ng/ml, and 200 ng/ml) on the growth of Ph⁺ B-ALL cells in the revised manuscript. Even at a low concentration of 25 ng/ml, IL-17A was found to promote the proliferation of Ph⁺ B-ALL cells (**Below panel, Fig 2d**).

Following other reviewers' suggestions, we also investigated the effect of pathological concentrations of IL-17A on the transcription of *IL-6*, *JAK2*, and *CXCL16* in B-ALL cells. Treatment with 200 pg/ml IL-17A increased the mRNA levels of *IL-6*, *JAK2*, and *CXCL16*, whereas treatment with 20-100 pg/ml IL-17A did not result in this effect (**Below panel, Fig 4h**). Additionally, we observed that treatment with pathological concentrations of IL-17A (20-500 pg/mL) significantly increased NF-κB activity (**Below panel, Fig 6g**) and the phosphorylation of BCR-ABL, STAT5, AKT, STAT3, p65 and MEK1/2 (**Below panel, Fig 4i**). These results suggest that IL-17A can activate the BCR-ABL, IL6/JAK/STAT3, and NF-κB signaling pathways in B-ALL cells at pathological concentrations ranging from 20 to 500 pg/mL.

19. Figure 5A: IL-17A staining seems absent. Why?

Response: The immunofluorescence staining image in the original Figure 5A is the merged image of DAPI, B220, CD4, and IL-17A staining. To improve the visualization of individual cell types in the spleen, we have presented the single staining images for DAPI (blue), B220 (lake blue), CD4 (green), and IL-17A (red) in the lower panel and Revised Fig 5a.

Fig 5a

20. Figure 5C: Please also use a BCR-ABL1-negative cell line.

Response: Following your suggestion, we determined the effect of IL-17A on chemokine expression in a Ph-negative leukemia cell line (NALM-6). CXCL16, CCL5, and IL-5 expression levels were increased after rhIL-17A treatment (Below panel, Supplementary Fig 5a). These results align with our previous observations in the Ph-positive leukemia cell lines SupB15 and BV173. However, the fold change in CXCL16 expression in NALM-6 cells treated with IL-17A (1-3-fold) was comparatively lower than those observed in SupB15 (9-14-fold) and BV173 (4-5-fold) cells. These data are shown in Revised Supplementary Fig 5a and Fig 5c.

Supplementary Fig 5a

Fig 5c

21. Figure 6C: For how long were cells treated?

Response: In Figure 6C, the cells used for immunofluorescence analysis were obtained from the spleens of WT mice and BCR-ABL^{tTA} mice. These mice were treated with or without rmlL-17A for three weeks. We have added detailed information to the legend of Figure 6c in the revised manuscript.

22. *Do Th17 cells express CXCR6??*

Response: The expression of CXCR6 was observed in Th17 cells, as demonstrated by Hou et al. (Hou and Yuki, 2022). In that study, the authors found that effector CD4 cells with a CCR6+CXCR6+ phenotype predominantly displayed classical Th17 markers such as IL23r, IL17A, and Rorc. Furthermore, it was found that CCR6 and CXCR6 can be used to identify cytotoxic Th17 cells in experimental autoimmune encephalomyelitis (Hou and Yuki, 2022). Therefore, it can be concluded that CXCR6 is indeed expressed in Th17 cells. We have added these descriptions to the Introduction section of the revised manuscript (page 4, lines 76-79).

Minor:

1. *The authors repeatedly speak of mouse models (e.g. p. 6 line 111), but at this point in the manuscript only one model has been tested.*

Response: Following your suggestion, we have modified the description of the mouse models in line 111 of the original manuscript to 'mouse model' in the revised manuscript (line 116 in the revised manuscript).

2. *The manuscript needs editing for English syntax and grammar.*

Response: Following your suggestion, we modified the revised manuscript for English syntax and grammar.

3. *More information is needed in the legends.*

Response: Following your suggestion, we have supplemented more detailed information to the figure legends in the revised manuscript.

4. *The authors repeatedly misuse the term 'xenotransplantation'. This only refers to transplantation between species.*

Response: Thank you for your professional suggestion. We have corrected the misused term 'xenotransplantation' in the revised manuscript.

5. *The discussion is largely a summary of the findings and needs to be improved.*

Response: Following your suggestion, we have revised our discussion section by adding new perspectives and deleting some summary findings in the revised manuscript.

6. Figure 2A: Schematic: Lymphocytes=Lymphoblasts

Response: Following your suggestion, we modified “Lymphocytes” to “Lymphoblasts” in the schematic representation in Revised Fig 2a and Fig 5i.

Fig 2a

Fig 5i

7. Quantifications of Western Blots are needed, as differences seem slight.

Response: Following your suggestion, we have quantified the indicated protein based on the band intensities from the Western blots in Revised Fig 4i, Fig 6e, and Fig 6i.

Fig 4i

Fig 6e

Fig 6i

Reviewer #2 (Remarks to the Author): with expertise in Th17/IL17 and hematological malignancies

Wang et al reported that the Th17 cells were enriched in niche microenvironment that promoted Ph+ B-ALL progression and increased CXCL16 expression in leukemia cells. They found that Th17 cells and Ph+ B-ALL cells promoted each other through IL-17A and CXCL16, forming a positive loop to enhance Ph+ B-ALL progression in the bone marrow niche. They further validated IL-17A or CXCL16 as a therapeutic target for treatment of Ph+ B-ALL alone or in combination with imatinib. The study was well designed and executed and the results are compelling. Overall finding is significant because the study provides the rationale and means to target IL-17A or CXCL16 for controlling B-ALL. However, there are several comments and questions for the authors to consider for further strengthening this excellent work. Specific comments are:

1) *Th17/IL-17 has been reported before by others to promote B cell lymphoma (e.g. Ferretti E et al, 2015; Bi L et al, 2016). The authors should cite the previously published studies and highlight the novelty of the current study.*

Response: Following your suggestion, we have cited the relevant references about the role of Th17/IL-17 in leukemia and lymphoma. Moreover, we highlighted the novelty of our study in the Discussion section of the revised manuscript (lines 337-345 and 400-407) as the following.

Studies have demonstrated that hematological malignancies, including multiple myeloma, B-cell lymphoma, AML, and ALL, exhibit elevated frequencies of Th17 cells (Bi et al., 2016; Han et al., 2014; Li et al., 2015; Prabhala et al., 2010; Tian et al., 2013). These increased numbers of Th17 cells and increased levels of IL-17 and other proinflammatory cytokines contribute to growth, drug resistance, and apoptosis inhibition in B-cell lymphoma cells and suppression of immune responses. These studies highlight the critical roles of Th17 cells and IL-17A in leukemia. However, determining whether Th17 cells could serve as a novel immunotherapy target to improve the outcomes of leukemia treatment remains challenging. Furthermore, the increased presence of Th17 cells in the leukemia niche and the relationship between IL-17A and leukemia progression *in vivo* requires further investigation.

Here, we found that Th17 cells are enriched in the niche microenvironment. The cytokine IL-17A, derived from Th17 cells, promotes Ph⁺ B-ALL progression and increases CXCL16 expression in leukemia cells. The increased CXCL16 further enhances Th17 cell differentiation and recruitment,

creating a positive loop that drives Ph⁺ B-ALL progression in the BM niche. Importantly, we further confirmed that targeting the Th17 niche or CXCL16 inhibits Ph⁺ B-ALL progression and exhibits a synergistic therapeutic effect in combination with imatinib *in vivo*. Our study provides valuable insights for developing a therapeutic approach combining anti-IL-17A or anti-CXCL16 neutralizing antibodies with the TKI imatinib to achieve complete remission of Ph⁺ B-ALL.

2) How Th17 cells were detected and defined should be clearly stated.

Response: Following your suggestion, we have shown the gating strategy of Th17 cells in Supplementary Fig 1a and provided a detailed description in the Methods section of the revised manuscript (page 27, lines 516-522).

Supplementary Fig 1a

3) IL-17A was used at 100ng/ml or even higher in the experiments. The concentrations are clearly beyond physiological or even pathological levels, which questions the relevance of the observed results. What is the range of IL-17A levels *in vivo* in mice or patients with B-ALL? Can IL-17A induce any signal sub nanogram levels?

Response: Thank you for your professional suggestion. We measured the plasma concentration of IL-17A in mice and patients with B-ALL. The plasma concentration of IL-17A ranged from 100 pg/ml to 500 pg/ml in *BCR-ABL^{TA}* mice and from 10 pg/ml to 50 pg/ml in patients with B-ALL (Below panel, Supplementary Fig 1j and Fig 1h). Th17 cells robustly accumulated near B-ALL cells in the spleen and BM (Fig 5a), which may result in the locally high concentration of IL-17A near B-ALL cells compared to the concentrations in plasma and liquid BM. However, it is difficult to accurately quantify the concentration of IL-17A in the immediate vicinity of B-ALL cells. In the original manuscript, we referred to published papers (Bi et al., 2016; Han et al., 2014; Li et al., 2015), in which concentrations of 50 ng/ml or 100 ng/ml IL-17A were used to stimulate leukemia cells. Therefore, the concentrations of IL-17A used in our original study were higher than pathological plasma levels. The high IL-17A concentration helps us to reach the platform or maximum effects of IL-17A in the system, which might mimic the niche microenvironments.

Following your suggestion, we investigated the effect of pathological concentrations of IL-17A on the transcription of *IL-6*, *JAK2*, and *CXCL16* in B-ALL cells. Treatment with 200 pg/ml IL-17A increased the mRNA levels of *IL-6*, *JAK2*, and *CXCL16*, whereas treatment with 20-100 pg/ml IL-17A did not result in this effect (Below panel, Fig 4h). Additionally, treatment with pathological concentrations of IL-17A (200-500 pg/mL) significantly increased the mRNA levels of *IL-6*, *JAK2*, and *CXCL16* (Below panel, Fig 4h); NF- κ B activity (Below panel, Fig 6g); and the phosphorylation of BCR-ABL, STAT5, AKT, STAT3, p65 and MEK1/2 (Below panel, Fig 4i). These results suggest that IL-17A can activate the BCR-ABL, IL6/JAK/STAT3, and NF- κ B signaling pathways in B-ALL cells at reasonable pathological concentrations (20-500 pg/mL).

4) Th17 cells also produce IL-21 and IL-22. Was either cytokine increased in B-ALL? If so, what is their contribution to B-ALL progress?

Response: Thank you for your professional suggestion. We measured the concentrations of IL-21 and IL-22 in the plasma of *BCR-ABL*^{tTA} B-ALL mice and *WT* mice. Although a slight increasing trend in plasma IL-21 and IL-22 concentrations was observed in *BCR-ABL*^{tTA} mice compared to *WT* mice, there was no statistically significant difference between these two groups (Below panel, Supplementary Fig 1k). Thus, we focused on exploring the role of IL17A, a cytokine secreted by Th17 cells that showed significant changes in the progression of Ph⁺ B-ALL.

Reviewer #3 (Remarks to the Author): with expertise in Th17/IL17 and hematological malignancies

In this manuscript by Wang et al, the authors systemically described a feed-back loop between Ph+ ALL and Th17 cells through the CXCL16/IL-17A axis which is accountable for adverse prognosis of this disease. Mechanistically, IL-17A promoted the progression of Ph+ ALL through activation of BCR-ABL, STAT3 and NF-kB pathways whereas inhibition of IL-17A attenuated the disease progression. The authors further demonstrated a feed-back loop between IL-17A and CXCL16 predominantly in vitro. More interestingly, inhibition of either IL-17A or CXCL16 was synergistic with imatinib for the treatment of Ph+ ALL. Although the correlation of IL-17A with disease progression in acute and chronic leukemia have been previously reported (in particular, Ref #22), the present study conducted deeper mechanistical investigations. Overall, this is a very interesting study and the findings are clinically translatable. However there are a few points that need further investigation or clarification.

Major comments:

1. The present study focuses on the interaction between Th17 cells (through IL-17A) and Ph+ ALL (CXCL16) which represents a direct crosstalk. There is no evidence or data that support a contribution from other factors in the microenvironment. Removal of “leukemia microenvironment” from the title can make the message of the story clearer.

Response: Thanks for your great suggestion. We removed “leukemia microenvironment” and modified the title to “Targeting IL-17A enhances imatinib efficacy in Philadelphia chromosome-positive B-cell acute lymphoblastic leukemia”.

2. The y-axis should be shown from 0 in all figures (for example 2C, 5G-J, 6B, 6J, 7A, 7E). The truncated y-axis exaggerates the difference and sometimes is misleading. For example, the difference between groups in Fig 6B/F are minimal (< 5 – 10%) but looks significantly bigger on the plots. As such, it's not convincing that the small changes in CXCL16 is accountable for the Th17 differentiation. In addition, the increase of CXCL16 in Ph+ ALL (Figure 6C) may reflect higher leukemia burden rather than an increase of expression on the per cell basis. What's the difference of systemic CXCL16 in the mouse model (IgG vs anti-IL-17A, WT vs BCR/ABL-tTA)?

Response: Following your suggestion, we have modified the y-axis presentation styles of all figures, including Fig 2c, Fig 5i, and Supplementary Fig 5b, Fig 5k, Fig 6b, Fig 6j, Fig 7a, Fig 7e, and Fig 7m.

Regarding the original Figures 6B and 6F, the IL-17A stimulation time of leukemia cells was 24 h, which may have been too short to induce CXCL16 secretion and NF-kB activation. To address this concern, we measured the CXCL16 concentration in the plasma of *BCR-ABL*^{tTA} B-ALL mice and WT mice. We observed that the concentration of CXCL16 in the plasma of *BCR-ABL*^{tTA} B-ALL mice (100-300 pg/ml) was increased at least 8-fold compared to that in *WT* mice (10-50 pg/ml) (Below panel, Fig 5e). Additionally, we performed an *in vitro* Th17 cell differentiation assay to investigate whether treatment with 200 pg/ml CXCL16 can affect the differentiation of naïve CD4+ T cells into Th17 cells. We did find that CXCL16 could induce Th17 cell differentiation at this concentration (Below panel, Supplementary Fig 5d).

In the original Figure 6C, we agree that the increase in CXCL16 in the spleen of Ph⁺ ALL mice might be due to the high leukemia burden, as you pointed out in this comment. Therefore, we have modified the description of Figure 6C in the revised manuscript (page 14, lines 280-283). In Revised Fig 6a and 6b, we found that IL-17A stimulation of primary B-ALL cells led to an increase in CXCL16 mRNA expression and CXCL16 protein secretion in Ph⁺ B-ALL cells, but only slight effects for the proliferation of these primary leukemia cells. Flow cytometric analysis showed CXCL16 expression in B-ALL cells after IL-17A treatment. The results revealed an increase in the mean fluorescence intensity of the CXCL16 signal in Ph⁺ B-ALL cells after IL-17A treatment, suggesting that IL-17A increases CXCL16 expression in Ph⁺ B-ALL cells on a per-cell basis (Below panel, Fig 6d).

Furthermore, we quantified the systemic CXCL16 level in *BCR-ABL*^{tTA} B-ALL mice treated with IgG or an anti-IL-17A neutralizing antibody. Compared to IgG treatment, treatment with the anti-IL-17A antibody significantly reduced the CXCL16 concentration in plasma (Below panel, Supplementary Fig 5b). These data, along with the data shown in Fig 6, indicate that IL-17A depletion attenuates the progression of Ph⁺ B-ALL by inhibiting the expression and secretion of CXCL16 in Ph⁺ B-ALL cells.

3. Please clarify the rationale to adopt xenograft model in Figure 7A-G. It's also very confusing that

rmCXCL16 is effective in this model since it doesn't have target in NSG mice. Please also describe the xenograft model in the methods section. What's the effect of rmCXCL16 treatment in BCR/ABL-tTA model which has the target?

Response: Thank you for your meticulous observation. We apologize for incorrectly describing the syngeneic transplantation mouse model as a xenograft model in the original manuscript. We performed secondary transplantation of *BCR-ABL^{tTA}* B-ALL cells into C57BL/6 mice to investigate the impact of CXCL16 on leukemogenesis *in vivo*. We have included a detailed description of the secondary transplantation of *BCR-ABL^{tTA}* B-ALL cells into C57BL/6 mice in the Methods section and revised the description of this model in the current manuscript (lines 454-466).

4. It's interesting to know if T cells (by depletion of CD4 or CD4/CD8 T cells) are beneficial or detrimental for the control of Ph+ ALL since T cells are significantly inhibited by chemotherapy and can be better targeted.

Response: Thank you for your suggestion. We previously transplanted Ph+ B-ALL cells derived from *BCR-ABL^{tTA}* mice into NOD-*scid*IL2Rg^{null} (NSG) mice with B and T-cell deficiency. These transplanted NSG mice died within ten days after transplantation, unlike transplanted immunocompetent mice (which died 30 days after transplantation). This suggests that B/T-cell deficiency promotes Ph+ ALL progression. Previous studies by Mumprecht et al., Oldreive et al., and Li et al. also support the hypothesis that lack of CD8+ T cells contributes to leukemia progression and Ph+ ALL relapse (Li et al., 2022b; Mumprecht et al., 2010; Oldreive et al., 2015).

CD4+ T cells are a heterogeneous population of cells that includes Th1, Th2, Th17, and Treg cells. Li et al. reported that depleting CD4+ T cells in Ph+ ALL mice did not affect the leukemia burden or survival rate but significantly increased disease relapse (Li et al., 2022b). However, whether deficiency of Th1, Th2, Th17, or Treg cells impacts Ph+ ALL is still unclear. In this study, we focused on investigating the role of elevated Th17 cells in Ph+ B-ALL and identifying potential therapeutic targets to increase the effectiveness of TKI (tyrosine kinase inhibitors) treatment. Further research is needed to understand the contribution of different CD4+ T-cell subpopulations to Ph+ ALL.

Minor comments

5. Ref #22 demonstrated that IL-17A promotes B-ALL proliferation by activating Akt/STAT3 (through IL-17RA). This should be cited and discussed in more detail.

Response: Following your suggestion, we have cited this reference and discussed it in the Discussion section (pages 19-20, lines 374-381) as follows: Bi et al. also reported that IL-17A promoted the proliferation of Ph- B-ALL cells through activation of the PI3K/AKT and Jak2/Stat3 signaling pathways (Bi et al., 2016). Here, we found that IL-17A treatment promoted the proliferation of Ph⁺ B-ALL cells by activating the IL6/JAK/STAT3 signaling pathway, consistent with the previous study. Additionally, we found that IL-17A treatment increased the activation of the BCR-ABL signaling pathway. The BCR-ABL oncoprotein is a critical molecular basis for leukemia pathogenesis, playing essential roles in cell proliferation, survival, and immunosuppression through the activation of several downstream signaling pathways, such as the JAK/STAT, PI3K/AKT, and Raf/MEK/ERK pathways (Naughton et al., 2009). Thus, targeting the IL-17A signaling pathway could be a valuable therapeutic approach for B-ALL.

6. The BCR/ABL-tTA mouse model should be described briefly in the results (or introduction) section.

Response: Following your suggestion, we have added the relevant data and described the BCR/ABL^{tTA} mouse model in the methods section of the revised manuscript as follows: tetO-BCR/ABL1 (B6.FVB/N-Tg(tetO-BCR/ABL1)2Dgt/Nju) mice (strain #N00005) were purchased from the Nanjing Biomedical Research Institute of Nanjing University (Nanjing, China). MMTV-tTA (B6.Cg-Tg(MMTVtTA)1Mam/J) mice (Strain #002618) were purchased from The Jackson Laboratory. C57BL/6J mice (8-10 weeks old) were purchased from Hua Fu Kang Technology Co., Ltd. (Beijing, China). BCR-ABL^{tTA} mice were generated by crossing female tetO-BCR/ABL1 (B6.FVB/N-Tg(tetO-BCR/ABL1)2Dgt/Nju) mice with male MMTV-tTA mice under continuous administration of tetracycline (0.5 g/L) in the drinking water. Withdrawal of tetracycline in BCR-ABL^{tTA} mice resulted in the development of Ph⁺ B-ALL within 1.5-2 months (Huettner et al., 2000).

Furthermore, we have provided detailed information about the BCR-ABL^{tTA} mice in the Results section (page 6, lines 94-106) of the revised manuscript as follows: Withdrawal of tetracycline administration in double transgenic mice (BCR-ABL^{tTA}; C57BL/6 (C57) background) induced the expression of BCR-ABL1 and resulted in the development of B-cell leukemia in 100% of the mice (Below panel, Supplementary Fig 1c). The survival time of BCR-ABL^{tTA} mice after tetracycline withdrawal was as long as 15 weeks (Below panel, Supplementary Fig 1d). Necropsy demonstrated massive splenomegaly and enlargement of the majority of lymph nodes (LNs) in these mice (Below panel, Supplementary Fig 1e-1f). FACS analysis revealed that the splenic lymphoblasts from the diseased mice were B220^{dim} (Below panel, Supplementary Fig 1g, left) and expressed CD19, CD43, and BP-1

(Below panel, Supplementary Fig 1g, right). This pattern suggested that the transformed cells underwent arrest at the pre-B-cell stage of development and was similar to the cell surface marker expression pattern identified in Ph⁺ B-ALL patient samples and MMTV-*BCR-ABL*^{tTA} transgenic mice (FVB background) (Harb et al., 2008; Hardy et al., 1991). Additionally, the frequency of immature blasts was found to be increased in Wright-Giemsa-stained peripheral blood (PB) and HE-stained BM and spleen sections of *BCR-ABL*^{tTA} mice compared to wild-type (WT) mice (Below panel, Supplementary Fig 1h).

Supplementary Fig 1c

Supplementary Fig 1d

Supplementary Fig 1e

Supplementary Fig 1f

Supplementary Fig 1g

Supplementary Fig 1h

7. Has CXCL16 been tested in the chemokine assay (Figure 1D)?

Response: CXCL16 was included in the chemokine assay, and its concentration was increased in the plasma of *BCR-ABL*^{tTA} B-ALL mice compared to WT mice.

8. TNF alpha should be written as TNF since TNF beta has been renamed.

Response: Following your suggestion, we have modified "TNF alpha" to "TNF" in the revised manuscript.

Reviewer #4 (Remarks to the Author): with expertise in B-ALL, tumor microenvironment

The authors describe a role for Th17 cells in promoting inflammation in the bone marrow of Ph+ B-ALL patients, and suggest blockade of IL-17 or of Th17 recruitment as a novel therapeutic strategy in B-ALL. While the idea of inflammation as a driver of B-ALL progression is interesting and important for the field, the article has several technical issues that have to be addressed to allow for correct interpretation of the role of Th17 cells in B-ALL.

1. In figure 1 (panels A-B), the authors claim that Th17 cells are increased in Ph+ B-ALL patients and mouse models, however it is unclear how the cells were detected (what gating strategy was used to identify Th17 cells?). Also, there are several single cell datasets from B-ALL published out there. Can the authors use them to further validate their Th17 findings?

Response: Following your suggestion, we have presented the gating strategy for identifying Th17 cells in the revised manuscript (Below panel, Supplementary Fig 1a and Fig 1a). Moreover, we analyzed the published single-cell datasets from healthy donors and Ph+ B-ALL patients (GSE134759). We found that the proportion of Th17 cells among the BMBCs from patients with Ph+ B-ALL was significantly higher than that from healthy donors (Below panel, Revised Fig 1d-1e). The hallmark “TNF signaling via NF-κB” pathway was also enriched in B cells from Ph+ B-ALL patients (Below panel, Revised Fig 4d).

2. In the cytokine assay (Fig 1D), the authors show an averaged column for all 3 WT mice and 3 separate columns for the 3 different BCR-ABL mice. It would be better to show separated data for the 3 WT mice as well. Also, showing z-scores would be more meaningful than showing averages, especially since it seems like all WT mice were normalized to 1.

Response: Thanks for your suggestion. Following the ideas from you and reviewer #1, we measured the concentrations of Th1/Th2/Th17 cytokines and chemokines in the plasma of additional BCR-ABL^{tTA} B-ALL mice and WT mice. The data were then analyzed by using z scores. A heatmap has been included in the revised manuscript to visually represent the z scores of the Th1/Th2/Th17 cytokines and chemokines in each sample. (Below panel, Fig 1f).

Fig 1f

3. In the Kaplan Meier curves in Fig 1H, it's not clear what the cutoff for high/low IL17A expression is – how were the groups determined?

Response: Thank you for your professional suggestion. The survival analysis in original Figure 1H (Revised Figure 1j) was performed using the “survminer” and “survival” R packages. The B-ALL samples were divided into groups based on the best separation cutoff value of IL-17A expression

(6.64, 35% of maximum IL-17A expression). Kaplan–Meier survival curves were plotted to compare the outcomes between the two groups. “IL-17A high” refers to the group of B-ALL patients with IL-17A mRNA expression above the cutoff value of 6.64, while “IL-17A low” refers to the group of B-ALL patients with IL-17A mRNA expression below the cutoff value of 6.64. We supplemented the related figure legend in the revised manuscript (page 46, lines 905-907).

4. In Figure 2, the authors demonstrate the effect of IL-17 on B-ALL cells. in panel E, the quantification of cells is unclear and it would be better to show bar plots with cell percentages for each condition/time. In 2I, the authors claim an effect for IL-17 in promoting B-ALL progression in vivo, here it would be good to show imaging data at an earlier time point, before IL-17 treatment started, to demonstrate that disease burden in the mice was equal before treatment. In the vehicle group, there are no signs of disease in 5 out of 8 mice at the time point shown, which could suggest a technical issue with engraftment. Similarly, the tissue infiltration and survival data (Fig 2K-L) could also be affected by this.

Response: Following the suggestion from you and Reviewer #1, we reconducted the engraftment experiment and assessed the leukemia burden at different time points (day 1, day 8, and day 15) in the revised manuscript (Below panel, Fig 2g). Biophotonic imaging on day 1 confirmed the successful transplantation of leukemia cells in all mice, with comparable leukemia burdens observed (Below panel, Fig 2h). By day 15, all mice transplanted with leukemia cells developed leukemia, indicating successful engraftment (Below panel, Fig 2h). The engraftment results showed that IL-17A treatment promoted Ph⁺ B-ALL progression (Below panel, Supplementary 2d), as indicated by the increased infiltration of leukemia cells in the PB, BM, and spleen (Below panel, Fig 2i- 2j), ultimately leading to a decreased overall survival rate in leukemia-engrafted mice (Below panel, Fig 2k).

5. In figure 4, the authors describe activation of inflammatory pathways in B-ALL cells following IL-

17A exposure. In Fig. 4A, it would be good to highlight on the volcano plot some of the genes upregulated or downregulated following IL-17A treatment.

Response: Following your suggestion, we highlighted some upregulated and downregulated genes following IL-17A exposure on the volcano plot. Inflammatory pathway-related genes, such as TNFRSF21, CXCL3, CX3CR1, and PTPN14, were upregulated following IL-17A treatment (Below panel, Fig 4a).

Fig 4a

6. The authors treat B-ALL cells with IL-17A and demonstrate activation of inflammatory cytokine transcription and of the JAK/STAT pathway. Here the authors use a high concentration of IL-17A (100ng/mL), whereas in the ELISA shown in figure 1 the concentrations measured in B-ALL patients were much lower (10-20 pg/mL). Would the same effect be seen when using physiological IL-17 concentrations? similarly in figure 6 the authors use very high concentrations of IL-17A. In addition, the western blot for pSTAT3 and p-p65 in panel 4F is hard to interpret – the authors should quantify the levels of pSTAT3 and p-p65 and perhaps use a shorter exposure for p-p65.

Response: Thank you for your professional suggestion. We measured the plasma concentration of IL-17A in mice and patients with B-ALL. The plasma concentration of IL-17A ranged from 100 pg/ml to 500 pg/ml in *BCR-ABL^{tTA}* mice and from 10 pg/ml to 50 pg/ml in patients with B-ALL (Below panel, Supplementary Fig 1j and Fig 1h). Th17 cells robustly accumulated near B-ALL cells in the spleen and BM (Fig 5a), which may result in the locally high concentration of IL-17A near B-ALL cells compared to the concentrations in the plasma and liquid BM. However, it is difficult to accurately quantify the concentration of IL-17A in the immediate vicinity of B-ALL cells. In the original manuscript, we referred to published papers (Bi et al., 2016; Han et al., 2014; Li et al., 2015), in which concentrations of 50 ng/ml or 100 ng/ml IL-17A were used to stimulate leukemia cells. Therefore, the concentrations of IL-17A used in our study were higher than pathological plasma levels. The high IL-

17A concentration just helps us to reach the platform or maximum effects of IL-17A in the system, which might mimic the niche microenvironments.

Following your suggestion, we investigated the effect of pathological concentrations of IL-17A on the transcription of *IL-6*, *JAK2*, and *CXCL16* in B-ALL cells. Treatment with 200 pg/ml IL-17A increased the mRNA levels of *IL-6*, *JAK2*, and *CXCL16*, whereas treatment with 20-100 pg/ml IL-17A did not result in this effect (Below panel, Fig 4h). Additionally, we observed that treatment with pathological concentrations of IL-17A (20-500 pg/mL) significantly increased the mRNA levels of *IL-6*, *JAK2*, and *CXCL16* (Below panel, Fig 4h); NF- κ B activity (Below panel, Fig 6g); and the phosphorylation of BCR-ABL, STAT5, AKT, STAT3, p65 and MEK1/2 (Below panel, Fig 4i). These results suggest that IL-17A can activate the BCR-ABL, IL6/JAK/STAT3, and NF- κ B signaling pathways in B-ALL cells at pathological concentrations (20-500 pg/mL). Moreover, we determined the effect of pathological concentrations of IL-17A (20-500 pg/mL) on p65 phosphorylation in the nucleus and cytoplasm. Treatment with pathological concentrations of IL-17A (20-500 pg/mL) significantly increased p65 phosphorylation in the nucleus in a dose-dependent manner in primary mouse B-ALL cells (Below panel, Fig 6e). In addition, we quantified the levels of p-STAT3 and p-p65 in Fig 4i.

7. In figure 5 the authors describe the crosstalk between B-ALL cells and Th17 cells. the migration assay shown in panel B is unclear – how was migration measured? Was the effect specific for Th17

cells, or were similar effects seen with other T cell subsets (CD4⁺/CD8⁺)? The authors use a B-ALL cell line, did the T cells used in the assay match the HLA of the cell line?

Response: Following your suggestion, we have added a detailed description of the migration assay in the Methods section of the revised manuscript (page 34, lines 672-677) as follows: For migration assays, Th17 cells were placed in the upper chamber containing a 5 µm pore polycarbonate membrane insert (Corning, 3421), and B-ALL cells were cultured in the lower chamber. After 12 h, the cells in the lower chamber were counted with a cell counter, and the percentage of CD4⁺ T cells in the lower chamber was determined by FCS analysis. The migrated Th17 cells in the lower chambers of the inserts were quantified as follows: cell number (lower chamber) x percentage of CD4⁺ T cells in the lower chamber.

To investigate whether other T-cell subsets (CD4⁺/CD8⁺) can be recruited by Ph⁺ B-ALL cells, we cocultured mouse T cells with primary mouse B-ALL cells in a noncontact coculture system (Below panel A). The T-cell migration assay showed that B-ALL cells could trigger CD4⁺ and CD8⁺ T-cell migration (Below panel B). We then further analyzed the CD4⁺ T-cell subsets whose migration was induced by B-ALL cells and found that the B-ALL cells could mediate the migration of Th2, Th17, and Treg cells (Below panel C). However, CD4⁺ and CD8⁺ T-cell infiltration was decreased in the spleens of *BCR-ABL*^{IT^A} B-ALL mice compared to those of WT mice (Below panel D), suggesting the regulatory complexity of T-cell subsets affected by B-ALL cells *in vivo*. In this study, we focused on exploring the effect of Th17 cells on the progression of B-ALL. Therefore, these data are shown below for your review and will not be presented in the revised manuscript.

Based on the HLA type information provided by ATCC, the HLA type of SupB15 cells does not match the HLA type of Th17 cells isolated from healthy donors. In the original MS, SupB15 cells were cocultured with Th17 cells in a noncontact coculture system. Our primary focus was exploring the chemokine-mediated recruitment of Th17 cells by B-ALL cells. However, we did not investigate the effector function of Th17 cells or leukemia proliferation due to the mismatch in HLA types within the coculture system.

We agree that using T cells with an HLA type matching primary B-ALL cells in this assay would be more appropriate. Therefore, we established a noncontact coculture system using Th17 cells derived from the same patient as the B-ALL cells. As expected, primary B-ALL cells successfully triggered the

migration of these Th17 cells, and this migration was further increased by rhIL-17A stimulation (**Below panel, Fig 5b**).

Fig. T-cell subsets recruited by Ph+ B-ALL cells (A) Schematic strategy for studying the migration of T cells cocultured with or without leukemia cells *in vitro*. Mouse T cells were isolated from the spleens of WT mice and then activated by anti-CD3, anti-CD28 or IL-2 treatment for 24 h. Then, the activated T cells were seeded in the upper chambers containing 5 μ m pore polycarbonate membrane inserts (Corning, 3421). Primary mouse B-ALL cells were seeded in the bottom chambers of the inserts. After 24 h of coculture, the cells in the bottom chambers of the inserts were counted using a cell counter and analyzed by FCS. **(B)** The numbers of CD4⁺ or CD8⁺ T cells that migrated in response to culture medium or leukemia cells were determined by cell counting and FCS analysis. The data are presented as the mean \pm S.E.M. of 3 independent experiments. Statistical significance was determined by a two-tailed Student's t-test. **(C)** The numbers of CD4⁺ T cell subsets (Th1, Th2, Th17 or Treg cells) that migrated in response to culture medium or leukemia cells were determined by cell counting and FCS analysis. The data are presented as the mean \pm S.E.M. of 3 independent experiments. Statistical significance was determined by a two-tailed Student's t-test. **(D)** Flow cytometric analysis of the percentage of CD4⁺ or CD8⁺ T cells in spleens isolated from WT mice or BCR-ABL^{tTA} mice (n=3 per group). The data are presented as the means \pm S.E.M. values. Statistical significance was determined by a two-tailed Student's t-test.

8. In figure 5C, the authors demonstrate an increase in transcription of several chemokines following IL-17A treatment – can the authors demonstrate secretion of these by ELISA?

Response: Following your suggestion, we assessed the impact of IL-17A on the secretion of CXCL5, IL-5, CXCL16, and CCL5. Our findings revealed significant increases in the secretion of IL-5, CXCL5, CCL5, and CXCL16 in SupB15 cells following rhIL-17A treatment (**Below panel, Fig 5d**).

9. The authors perform co-culture of T cells derived from healthy donors with primary B-ALL samples – were the HLA isotypes matched for these assays? Can the authors perform the same experiment with T cells derived from the same patient as the tumor cells?

Response: The HLA types of T cells isolated from healthy donors were haploidentical with those of the primary B-ALL samples used in the original manuscript. Following your suggestion, we isolated Th17 cells and leukemia cells from the same patients. Adding Th17 cells to the coculture increased the percentage of Ki-67⁺ leukemia cells and increased the survival of Ph⁺ B-ALL cells. However, this effect was blocked when an anti-IL-17A neutralizing antibody was added to the coculture system (Below panel, Fig 2a-2c).

Furthermore, we evaluated the effect of the anti-CXCL16 neutralizing antibody on the migration of Th17 cells and the proliferation activity of leukemia cells in this noncontact coculture system using Th17 cells and leukemia cells isolated from the same patients (Below panel, Fig 5i). Anti-CXCL16 treatment significantly inhibited the migration of Th17 cells (Below panel, Fig 5j) and suppressed the proliferation of leukemia cells (Below panel, Fig 5k).

Fig 2a**Fig 2b****Fig 2c****Fig 5i****Fig 5j****Fig 5k**
Reference

- Bi, L., Wu, J., Ye, A., Wu, J., Yu, K., Zhang, S., and Han, Y. (2016). Increased Th17 cells and IL-17A exist in patients with B cell acute lymphoblastic leukemia and promote proliferation and resistance to daunorubicin through activation of Akt signaling. *J Transl Med* *14*, 132.
- Chen, J., Liao, M.Y., Gao, X.L., Zhong, Q., Tang, T.T., Yu, X., Liao, Y.H., and Cheng, X. (2013). IL-17A induces pro-inflammatory cytokines production in macrophages via MAPKs, NF-kappaB and AP-1. *Cell Physiol Biochem* *32*, 1265-1274.
- Guery, L., and Hugues, S. (2015). Th17 Cell Plasticity and Functions in Cancer Immunity. *Biomed Res Int* *2015*, 314620.
- Han, Y., Ye, A., Bi, L., Wu, J., Yu, K., and Zhang, S. (2014). Th17 cells and interleukin-17 increase with poor prognosis in patients with acute myeloid leukemia. *Cancer Sci* *105*, 933-942.
- Harb, J.G., Chyla, B.I., and Huettner, C.S. (2008). Loss of Bcl-x in Ph+ B-ALL increases cellular proliferation and does not inhibit leukemogenesis. *Blood* *111*, 3760-3769.
- Hardy, R.R., Carmack, C.E., Shinton, S.A., Kemp, J.D., and Hayakawa, K. (1991). Resolution and characterization of pro-B and pre-pro-B cell stages in normal mouse bone marrow. *J Exp Med* *173*, 1213-1225.

Hou, L., and Yuki, K. (2022). CCR6 and CXCR6 Identify the Th17 Cells With Cytotoxicity in Experimental Autoimmune Encephalomyelitis. *Front Immunol* *13*, 819224.

Huettner, C.S., Zhang, P., Van Etten, R.A., and Tenen, D.G. (2000). Reversibility of acute B-cell leukaemia induced by BCR-ABL1. *Nat Genet* *24*, 57-60.

Li, Q., Xu, X., Zhong, W., Du, Q., Yu, B., and Xiong, H. (2015). IL-17 induces radiation resistance of B lymphoma cells by suppressing p53 expression and thereby inhibiting irradiation-triggered apoptosis. *Cell Mol Immunol* *12*, 366-372.

Li, Y., Chang, L.H., Huang, W.Q., Bao, H.W., Li, X., Chen, X.H., Wu, H.T., Yao, Z.Z., Huang, Z.Z., Weinberg, S.E., *et al.* (2022a). IL-17A mediates pyroptosis via the ERK pathway and contributes to steroid resistance in CRSwNP. *J Allergy Clin Immunol* *150*, 337-351.

Li, Y., Yang, X., Sun, Y., Li, Z., Yang, W., Ju, B., Easton, J., Pei, D., Cheng, C., Lee, S., *et al.* (2022b). Impact of T-cell immunity on chemotherapy response in childhood acute lymphoblastic leukemia. *Blood* *140*, 1507-1521.

Mathisen, M.S., O'Brien, S., Thomas, D., Cortes, J., Kantarjian, H., and Ravandi, F. (2011). Role of tyrosine kinase inhibitors in the management of Philadelphia chromosome-positive acute lymphoblastic leukemia. *Curr Hematol Malig Rep* *6*, 187-194.

Mazzera, L., Abeltino, M., Lombardi, G., Cantoni, A.M., Jottini, S., Corradi, A., Ricca, M., Rossetti, E., Armando, F., Peli, A., *et al.* (2023). MEK1/2 regulate normal BCR and ABL1 tumor-suppressor functions to dictate ATO response in TKI-resistant Ph+ leukemia. *Leukemia*.

Mombelli, S., Cochaud, S., Merrouche, Y., Garbar, C., Antonicelli, F., Laprevotte, E., Alberici, G., Bonnefoy, N., Eliaou, J.F., Bastid, J., *et al.* (2015). IL-17A and its homologs IL-25/IL-17E recruit the c-RAF/S6 kinase pathway and the generation of pro-oncogenic LMW-E in breast cancer cells. *Sci Rep* *5*, 11874.

Mumprecht, S., Schurch, C., Scherrer, S., Claus, C., and Ochsenbein, A.F. (2010). Chronic myelogenous leukemia maintains specific CD8(+) T cells through IL-7 signaling. *Eur J Immunol* *40*, 2720-2730.

Najafi, S., and Mirshafiey, A. (2019). The role of T helper 17 and regulatory T cells in tumor microenvironment. *Immunopharmacol Immunotoxicol* *41*, 16-24.

Naughton, R., Quiney, C., Turner, S.D., and Cotter, T.G. (2009). Bcr-Abl-mediated redox regulation of the PI3K/AKT pathway. *Leukemia* *23*, 1432-1440.

Oldreive, C.E., Skowronska, A., Davies, N.J., Parry, H., Agathangelou, A., Krysov, S., Packham, G., Rudzki, Z., Cronin, L., Vrzalikova, K., *et al.* (2015). T-cell number and subtype influence the disease course of primary chronic lymphocytic leukaemia xenografts in alymphoid mice. *Dis Model Mech* *8*, 1401-1412.

Onishi, R.M., and Gaffen, S.L. (2010). Interleukin-17 and its target genes: mechanisms of interleukin-17 function in disease. *Immunology* *129*, 311-321.

Pesce, B., Ribeiro, C.H., Larrondo, M., Ramos, V., Soto, L., Catalan, D., and Aguillon, J.C. (2022). TNF-alpha Affects Signature Cytokines of Th1 and Th17 T Cell Subsets through Differential Actions on TNFR1 and TNFR2. *Int J Mol Sci* 23.

Prabhala, R.H., Pelluru, D., Fulciniti, M., Prabhala, H.K., Nanjappa, P., Song, W., Pai, C., Amin, S., Tai, Y.T., Richardson, P.G., *et al.* (2010). Elevated IL-17 produced by TH17 cells promotes myeloma cell growth and inhibits immune function in multiple myeloma. *Blood* 115, 5385-5392.

Salazar, Y., Zheng, X., Brunn, D., Raifer, H., Picard, F., Zhang, Y., Winter, H., Guenther, S., Weigert, A., Weigmann, B., *et al.* (2020). Microenvironmental Th9 and Th17 lymphocytes induce metastatic spreading in lung cancer. *J Clin Invest* 130, 3560-3575.

Tian, T., Sun, Y., Li, M., He, N., Yuan, C., Yu, S., Wang, M., Ji, C., and Ma, D. (2013). Increased Th22 cells as well as Th17 cells in patients with adult T-cell acute lymphoblastic leukemia. *Clin Chim Acta* 426, 108-113.

Xing, X., Yang, J., Yang, X., Wei, Y., Zhu, L., Gao, D., and Li, M. (2013). IL-17A induces endothelial inflammation in systemic sclerosis via the ERK signaling pathway. *PLoS One* 8, e85032.

REVIEWERS' COMMENTS

Reviewer #1 (Remarks to the Author):

In this revised version of the manuscript the authors have made significant improvements. Most concerns have been addressed, and the authors describe the complex network of cytokines in the tumor microenvironment of B-ALL.

Further comments are:

1. The results of the homing experiments clearly show that homing is an important factor in the authors' findings. This should be added to the discussion.
2. The amount of data on clinical samples is fairly small. It is advisable to add some text about the relevance of their findings to the human setting in the discussion.
3. How do the authors explain the difference between Figures 1g and 1i in the IL-17RA groups (lower in g, higher in i)?

Reviewer #2 (Remarks to the Author):

My previous comments are adequately addressed. No further comments,

Reviewer #3 (Remarks to the Author):

All of my major concerns have been properly addressed. There is one minor comment (see below).

Please clarify "xenograft" in line 310 (Figure 7a) which generally refers to transplantation across different species. It would be more appropriate to use "syngeneic transplant" or "isograft".

Reviewer #4 (Remarks to the Author):

The authors made a significant effort to respond to all comments of the reviewers. Their rebuttal letter was impressive and addressed all our criticisms. This is a very exciting finding

that i believe will add something significant to our understanding of the interaction of leukemia with its immune microenvironment.

Reviewer #1 (Remarks to the Author):

In this revised version of the manuscript the authors have made significant improvements. Most concerns have been addressed, and the authors describe the complex network of cytokines in the tumor microenvironment of B-ALL.

Further comments are:

1. The results of the homing experiments clearly show that homing is an important factor in the authors' findings. This should be added to the discussion.

Re: Thanks again for your professional suggestion about homing experiments. Salvestrini et al. reported that the leukemia cells homing to the bone marrow niche resist chemotherapeutic drugs (Salvestrini et al., 2012). In this study, we found that IL-17A promoted but anti-IL-17A monoclonal antibodies (mAbs) treatment reduced leukemia cells' homing, suggesting that anti-IL-17A mAbs treatment may render leukemia cells more sensitive to standard chemotherapeutic drugs. We added these comments in the discussion section.

2. The amount of data on clinical samples is fairly small. It is advisable to add some text about the relevance of their findings to the human setting in the discussion.

Re: Thanks for this great suggestion. Monoclonal antibodies targeting IL-17A, including Secukinumab and Ixekizumab, have been approved to treat moderate to severe plaque psoriasis (McInnes et al., 2015; Toussirot, 2018). Our study suggests anti-IL-17A antibodies may achieve significant therapeutic efficacy in Ph⁺ B-ALL patients under appropriate conditions, especially with TKI therapy. We added these comments in the discussion section.

3. How do the authors explain the difference between Figures 1g and 1i in the IL-17RA groups (lower in g, higher in i)?

Re: We thank you for pointing out this mRNA and protein discrepancy. After we noticed the Th17 cell population enrichment and higher IL-17A expression (Figure 1f) in Ph⁺ B-ALL patients or *BCR-ABL*^{T7A} mice, we wanted to address the expression of IL-17RA/IL-17RC receptor complex. In Figure 1g, we queried the GEO database about

IL-17 RA mRNA expression in Ph⁺ B-ALL. Although the mRNA expression of *IL-17 RA* in Ph⁺ B-ALL patients was lower than that in Healthy Donors (Figure 1g), the protein levels of IL-17RA remained similar between Ph⁺ B-ALL patients and Healthy Donors (Figure 1i). Moreover, we confirmed higher IL-17RC expression in Ph⁺ B-ALL patients than in Healthy Donors (Figure 1i). Although the sample size of our clinical samples (Figure 1i) was relatively small compared to the sample size from the GEO database (Figure 1g), these results indicated that the presence of IL-17RA and IL-17RC on the leukemia cell surface of primary Ph⁺ B-ALL cells. The similar IL-17RA expression and higher IL-17RC expression enable the IL-17A downstream signaling in leukemia cells. We revised our results section and tried to make it clear.

Reviewer #2 (Remarks to the Author):

My previous comments are adequately addressed. No further comments,

Re: We thank your constructive comments regarding our study.

Reviewer #3 (Remarks to the Author):

All of my major concerns have been properly addressed. There is one minor comment (see below).

Please clarify "xenograft" in line 310 (Figure 7a) which generally refers to transplantation across different species. It would be more appropriate to use "syngeneic transplant" or "isograft".

Re: Following your professional suggestion, we have modified the "xenograft" in line 310 to "syngeneic transplant"(Figure 7a).

Reviewer #4 (Remarks to the Author):

The authors made a significant effort to respond to all comments of the reviewers. Their rebuttal letter was impressive and addressed all our criticisms. This is a very exciting finding that I believe will add something significant to our understanding of the interaction of leukemia with its immune microenvironment.

Re: Thanks again for your constructive comments on our manuscript.

Reference

Salvestrini, V., Zini, R., Rossi, L., Gulinelli, S., Manfredini, R., Bianchi, E., Piacibello, W., Caione, L., Migliardi, G., Ricciardi, M.R., *et al.* (2012). Purinergic signaling inhibits human acute myeloblastic leukemia cell proliferation, migration, and engraftment in immunodeficient mice. *Blood* 119, 217-226.

McInnes, I.B., Mease, P.J., Kirkham, B., Kavanaugh, A., Ritchlin, C.T., Rahman, P., van der Heijde, D., Landewe, R., Conaghan, P.G., Gottlieb, A.B., *et al.* (2015). Secukinumab, a human anti-interleukin-17A monoclonal antibody, in patients with psoriatic arthritis (FUTURE 2): a randomised, double-blind, placebo-controlled, phase 3 trial. *Lancet* 386, 1137-1146.

Toussiot, E. (2018). Ixekizumab: an anti-IL-17A monoclonal antibody for the treatment of psoriatic arthritis. *Expert Opin Biol Ther* 18, 101-107.